# Fate specification is spatially intermingled across planarian stem cells

Chanyoung Park [1,2,5], Kwadwo E. Owusu-Boaitey[1,2,3,5], Giselle M. Valdes [1,2] & Peter W. Reddien [1,2,4] ✉

Regeneration requires mechanisms for producing a wide array of cell types. Neoblasts are stem cells in the planarian *Schmidtea mediterranea* that undergo fate specification to produce over 125 adult cell types. Fate specification in neoblasts can be regulated through expression of fate-specific transcription factors. We utilize multiplexed error-robust fluorescence in situ hybridization (MERFISH) and whole-mount FISH to characterize fate choice distribution of stem cells within planarians. Fate choices are often made distant from target tissues and in a highly intermingled manner, with neighboring neoblasts frequently making divergent fate choices for tissues of different location and function. We propose that pattern formation is driven primarily by the migratory assortment of progenitors from mixed and spatially distributed fate-specified stem cells and that fate choice involves stem-cell intrinsic processes.

In adult regeneration, stem and progenitor cells must correctly choose their fates among a myriad of options. The mechanistic framework for fate specification is largely established from studies of development, where lineage, position, and local cues are major influencing factors[1–7]. Adult regeneration, however, presents distinct challenges that could have major impact on fate specification mechanisms. First, regeneration must respond to a wide variety of potential initiation points with different missing tissue types and amounts. Second, fate specification in regeneration for some species occurs in the context of differentiated tissues, presenting a different cell communication environment. Third, regeneration can lack development-specific stages that facilitate the initiation and refinement of fate choice. Finally, regeneration in some species can initiate from abundant progenitors, whereas substantial progenitor amplification must often occur prior to differentiation in development.

The planarian *Schmidtea mediterranea* presents an attractive system for study of fate specification in adult stem cells[8]. Planarians possess complex anatomy including a brain containing diverse neuronal cell types, eyes, musculature, intestine, protonephridia (an excretory and osmoregulatory system), phagocytic cells, and epidermis. Planarians are highly regenerative, able to restore any missing

body part and even to regenerate a complete animal from a small fragment. This ability is driven by a pluripotent population of adult stem cells called neoblasts, which produce all of the 125+ adult planarian cell types in regeneration and in continual tissue turnover[9–14]. Neoblasts are the only dividing somatic cells in adult planarians, and transplant of a single neoblast is sufficient to rescue regeneration in animals lacking neoblasts[12]. Neoblast proliferation at wound sites generates an outgrowth called a blastema, in which missing tissues can differentiate.

Fate specification prominently occurs in neoblasts through the expression of fate-specific transcription factors (FSTFs), generating specialized neoblasts[15–18]. Many FSTFs are required for specialized neoblasts to produce their target differentiated cell types across major adult tissues. A large proportion of S/G2/M phase neoblasts express FSTFs, and fate specification from naive to post-mitotic differentiating state can occur within a single cell cycle[19]. Specialized neoblasts can divide to produce an asymmetric outcome, with a post-mitotic daughter of a single fate and a neoblast daughter; this neoblast daughter can then itself divide and choose a new fate[19].

Extensive work has uncovered FSTFs and specialized neoblasts for all major tissue classes in planarians[18]. Some specialized neoblasts are

[1]Whitehead Institute for Biomedical Research, Cambridge, MA, USA. [2]Department of Biology, Massachusetts Institute of Technology, Cambridge, MA, USA. [3]Harvard/MIT MD-PhD Program, Harvard Medical School, Boston, MA, USA. [4]Howard Hughes Medical Institute, Chevy Chase, MD, USA. [5]These authors contributed equally: Chanyoung Park, Kwadwo E. Owusu-Boaitey. ✉e-mail: reddien@wi.mit.edu

distributed throughout the parenchymal space in spatially broad domains, such as for the epidermis[20]. Certain fates are specified in a more spatially restricted manner, such as for the eye, which are specified in the head[21]. However, the specification domains of regionally restricted fates such as the eye are also broad[21]. The mechanisms by which neoblasts correctly choose from over 125 transcriptionally distinct mature cell fates remains poorly understood.

## Results

### MERFISH reveals the spatial organization of distinct cell types

To understand the processes underlying neoblast fate choice, we first sought to spatially map specialized neoblast distributions. Multiplexed error-robust fluorescence in situ hybridization (MERFISH) is a single-molecule FISH methodology that can detect hundreds of distinct RNA species with subcellular resolution in tissue sections[22]. MERFISH with tissue-specific probes robustly and simultaneously labeled all major planarian tissue classes, including the intestine (*mat*), epidermis (*PRSS12*), muscle (*colF-2*), protonephridia (*cubilin*), cholinergic neurons (*ChAT*), pigment cells (*pbgd*), pharynx (*VIT*), pharyngeal phagocytic/*cathepsin*+ cells (*TTPA*), parenchymal cell types (dd4761; *fer3l-2; ZAN6; glipr-1;* dd829; *X1.A.B7.1; mag-1; FAM115C-like;* dd385; *SSPO*), and

neoblasts (*smedwi-1; bruli; soxP-1; vasa-1; mcm-7; soxP-2; rtel-1; znf333; nanos; fgfr-4; smedwi-2; H2B; fgfr-1*) (Fig. 1a, Supplementary Fig. 1, Supplementary Table 1). Distinct cell populations of mature cell types within the parenchymal tissue (Supplementary Fig. 2), protonephridia (Supplementary Fig. 3), nervous system (Supplementary Fig. 4), and pharynx (Supplementary Fig. 5) were also readily labeled. Cell-type classification was facilitated by assessment of the pooled expression of multiple cell-type enriched markers (Supplementary Fig. 6a). Neoblasts were classified based on high expression of *smedwi-1* and expression of additional known genes with neoblast-enriched expression (Supplementary Fig. 6b, c). MERFISH presents a powerful new methodology to analyze the spatial organization of planarian cell types.

To systematically characterize the spatial distribution of stem cell fate choices, we studied 61 FSTFs. For some fates we utilized FSTF fate expression signatures to improve the detection sensitivity and specificity of specialized neoblast types (Supplementary Table 2). Pooled FSTF expression signatures were used to identify epidermal, muscle, intestinal, neural, eye, and protonephridial specialized neoblasts (Supplementary Figs. 7–10). Neoblasts without clear fate signatures were unassigned, and could reflect G1 neoblasts, neoblasts positive for

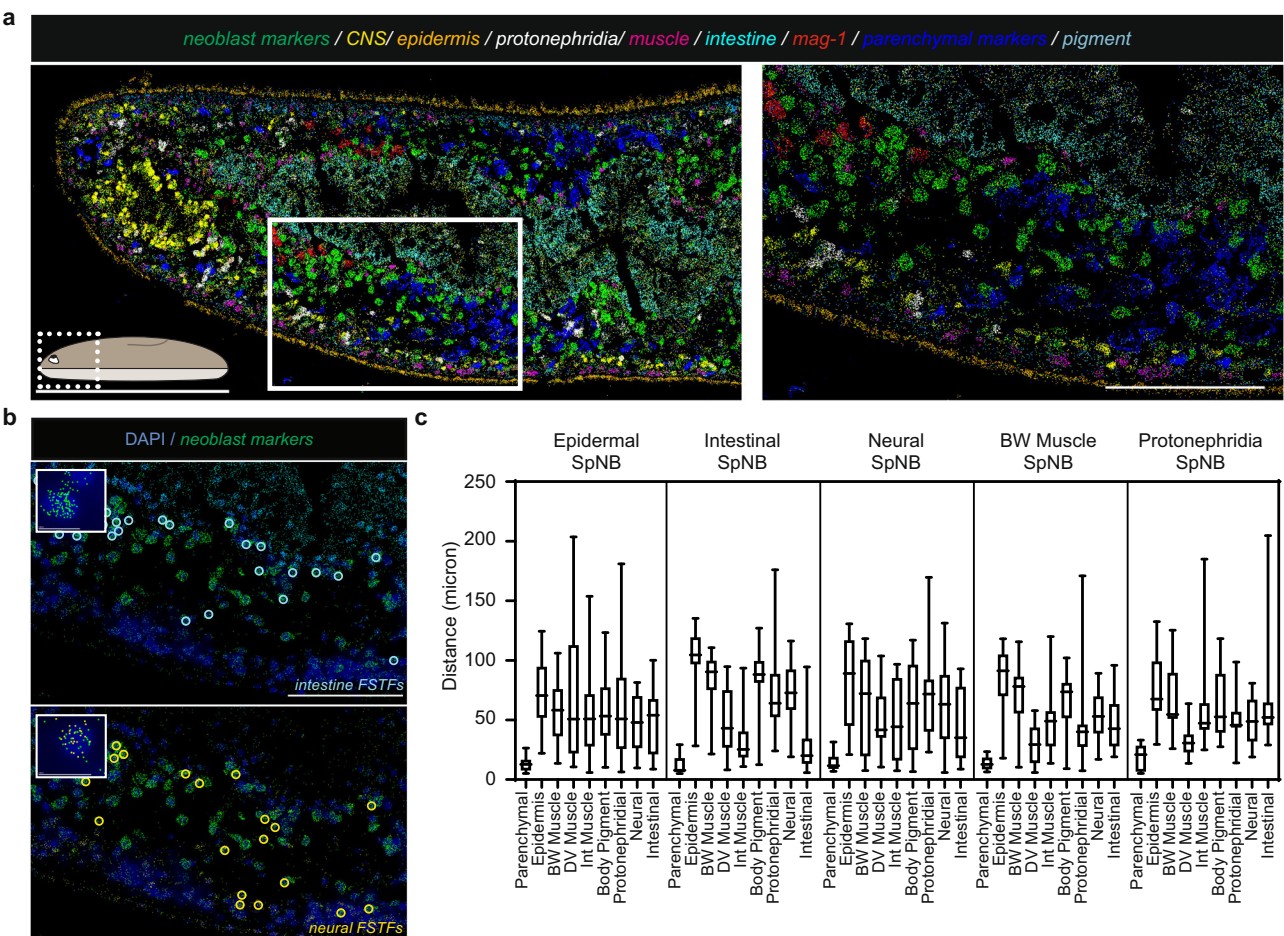

**Fig. 1 | Simultaneous detection of numerous differentiated cells and specialized neoblast classes by MERFISH. a** MERFISH labeling of major tissue classes in a thin sagittal section. Cartoon represents spatial location of depicted region. Scale bar, 250 μm; inset region (right), 100 μm. **b** Identification of specialized neoblasts by MERFISH. Colored circles represent identified specialized neoblasts. Scale bar, 100 μm. Inset depicts a representative detected specialized neoblast. Images represent fate signatures comprised of multiple gene targets described in Supplementary Table 2. Scale bar, 10 μm. Representative images from a single tissue section are shown in **a** and **b** from n = 3 animals. **c** Distances of identified specialized

neoblasts to nearest mature tissues. Each condition contains measurements from specialized neoblasts of a given class within the region depicted in **b**, except for protonephridial specialized neoblasts which were measured in nearby anatomical regions. The upper and lower hinges correspond to 25th and 75th percentiles, whiskers represent the smallest and largest values, and the line corresponds to the median. n = 17 (Epidermal SpNB), 27 (Intestinal SpNB), 20 (Neural SpNB), 15 (BW Muscle SpNB), 11 (Protonephridial SpNB). SpNB, specialized neoblasts. Source data are provided as a Source Data file.

untested FSTFs, or unspecialized neoblasts (Supplementary Figs. 8–10). Whereas transcripts for some FSTFs were highly specific to a particular specialized class, others were enriched but not exclusive to a single class (Supplementary Figs. 8–10). These observations are consistent with previously reported single cell RNA-sequencing data, and could reflect roles in multiple states or technical/biological noise[13]. For example, *ovo* transcripts are specifically present in eye-specialized neoblasts but *zfp-1* transcripts are enriched in epidermal specialized neoblasts and also present in some intestinal specialized neoblasts to a lower degree[13]. In such cases, specialized neoblasts were classified by a clear enrichment for a signature of transcripts indicative of a particular single fate class (Supplementary Figs. 8–10).

Multiple specialized neoblast classes of distinct fates were identified within the same animal regions and without overt spatial organization (Fig. 1b, Supplementary Fig. 11a). Specialized neoblasts were distributed throughout the mesenchymal space; for example, intestinal, neural, muscle, and epidermal specialized neoblasts, as well as unassigned neoblasts, spanned from near the animal surface to the innermost region adjacent to the intestine (Fig. 1b, Supplementary Fig. 11a, b). Specialized neoblasts were, in many cases, as close or closer to multiple non-target differentiated tissues than to target tissues (Fig. 1c, Supplementary Fig. 11c, d). For example, specialized neoblasts for body-wall muscle were on average 72 microns (-7 neoblast cell diameters) away from body-wall muscle, closer on average to parenchymal cells than to the body-wall muscle itself, and a similar average distance to multiple other differentiated tissues (Fig. 1c, Supplementary Fig. 11d). Furthermore, the variance of the mean distance of specialized neoblasts to their target tissue was large, indicating a lack of precise localization for these cells, which instead were spread across the mesenchymal space (Fig. 1c). These attributes were apparent for all tested specialized neoblast classes. In every case, specialized neoblasts were on average much closer to parenchymal cells, a tissue that includes gland cells, than to their target tissue (Supplementary Fig. 11d). Parenchymal cells were in fact nestled in close proximity with neoblasts of many classes, frequently representing the closest neighbor cell (Supplementary Fig. 11e). This physical association is notable, but no regulatory role for parenchymal cells in neoblast biology is as of yet known.

Planarians possess a complex nervous system comprised of many transcriptionally distinct cell types[13]. Serotonergic neurons express *sert* and *tph,* and are generated by specialized neoblasts that express *pitx*[23,24]. *sert*+ serotonergic neurons and *pitx*+ specialized neoblasts were robustly detected in MERFISH experiments (Supplementary Fig. 12a–c). *pitx*+ specialized neoblasts were closer on average to parenchymal cell types than to other mature cell types (Supplementary Fig. 12d, e). Serotonergic neural specialized neoblasts were also not positioned closer to mature serotonergic neurons than to other mature neuron types, indicating that these fates were not specified directly adjacent to their target tissue (Supplementary Fig. 12f). Similarly, dorsally located neural specialized neoblasts were on average a similar distance to multiple differentiated tissues and closer to parenchymal cell types than to dorsal mature neural cell types, consistent with data from ventrally assessed neural specialized neoblast populations (Fig. 1c, Supplementary Fig. 12g, h).

Intestinal specialized neoblasts were on average closer to the intestine than to body-wall tissues. However, these cells also displayed large variability in their distance to their target tissue, indicative of a distributed spatial organization (Fig. 1c). Three-dimensional mapping of body-wall muscle and intestinal specialized neoblasts using whole-mount FISH corroborated these results (Supplementary Fig. 13). However, the intestine-adjacent mesenchyme was not exclusively occupied by intestinal specialized neoblasts; for example, many body-wall muscle-specialized neoblasts were located at similar distances from the intestine (Supplementary Fig. 13b, c). Similarly, intestinal and epidermal specialized neoblasts were present in overlapping spatial

regions in sagittal sections (Supplementary Fig. 14). Whereas it is possible that neoblasts undergoing fate specification might exhibit some movement towards their target tissues, previous work indicates that neoblasts are not highly mobile under homeostatic conditions[25,26]. Thus, the specification of some fates was biased towards their target tissue, but occurred broadly and in overlapping fate-specification domains for distinct cell types.

## Specialized neoblasts of the same fate do not cluster together

We considered the possibility that particular fate decisions would display a locally clustered pattern. This possibility was examined in three dimensions using whole-mount FISH for numerous neoblast fates (Supplementary Table 3). The nearest neighbor neoblast of a given specialized neoblast was frequently not of that class ("non-self", Fig. 2a, Supplementary Fig. 15, Supplementary Movie 1). This scattered neoblast-specialization pattern occurred in different regions and across animal body axes for multiple fate classes (Fig. 2b, Supplementary Figs. 16–18). Epidermal, muscle, and protonephridial specialized neoblasts were significantly closer to non-self (tested FSTF⁻) neoblasts than to self-class (tested FSTF⁺) specialized neoblasts (Fig. 2c). Furthermore, the nearest neighbor neoblast for all tested fates was non-self the majority of the time (Fig. 2d). The observed proportions of same-class nearest neighbors were somewhat higher than that expected from random sampling of neoblasts from their overall proportion in all neoblasts (Fig. 2b–d, Supplementary Fig. 18c). This is consistent with some bias of specialized neoblast location toward their target tissue described above. Recent work demonstrated that FSTF activation in S-phase neoblasts marks the initial time of neoblast fate choice, and that a majority of S/G2/M neoblasts express FSTFs[19]. This suggests that cells in S-phase mark the location of cells when choice is made. We treated animals with a short (8 h) F-ara-EdU (EdU) pulse to mark neoblasts replicating DNA (Supplementary Fig. 19a); EdU⁺ specialized neoblasts were frequently surrounded by non-self, EdU⁺ cells, corroborating the interpretation that initial fate choices are not made in distinct clusters (Supplementary Fig. 19b).

## Specialized neoblasts are frequently neighbored by neoblasts of a different fate

To assess whether different fates can be specified in close proximity, we performed combinatorial whole-mount FISH and mapped the three-dimensional positions of specialized neoblast pairs (Fig. 3, Supplementary Movies 2–5). Eye-specialized neoblasts were intermingled among non-self neoblasts and interspersed by numerous neoblasts choosing epidermal or intestinal fates (Fig. 3a). Similarly, muscle-specialized and protonephridial specialized neoblasts were interspersed with epidermal specialized neoblasts (Fig. 3b, Supplementary Movies 6, 7).

We systematically examined immediate neoblast proximity in three dimensions with pair-wise tests for epidermal, intestinal, body-wall muscle, neural, protonephridial, pharyngeal, and eye fates (Fig. 3c, Supplementary Fig. 20). Strikingly, fate specification for tissues typically considered to be derived from different embryonic germ layers across diverse animal embryos frequently occurred in directly adjacent neoblasts (Fig. 3c, Supplementary Figs. 20 and 21, Supplementary Movie 8). This association was also visible by MERFISH using single markers (Supplementary Fig. 20b). For example, neoblasts with fate for the typically endoderm-derived intestine were present next to neighboring neoblasts with fate for typically ectoderm-derived epidermis (Fig. 3c, Supplementary Fig. 21). Thus, the pattern of fate specification in planarian stem cells deviates substantially from the spatial organization of fate specification in most animal embryos. Additional species capable of whole-body regeneration possess pluripotent or multipotent adult stem cells, and it will be of interest to determine how fate choices associated with different germ layers are made in these contexts[27]. Neighboring pairs of disparate fates were found for every

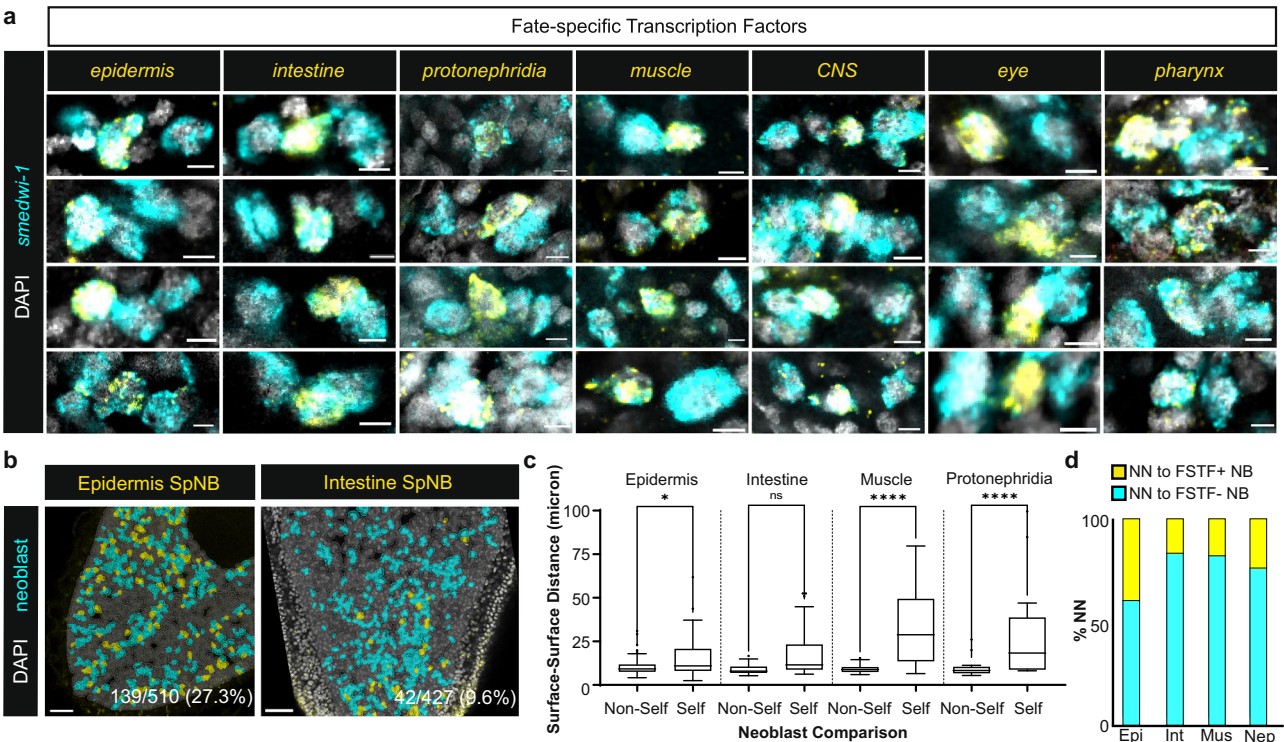

**Fig. 2 | Fate choices are not made in spatially distinct clusters. a** Specialized neoblasts (SpNBs) are often in proximity to non-self neoblasts. Genes for tissue-specific FSTF pools: epidermis (*soxP-3*), intestine (*hnf-4, gata4/5/6-1*), protonephridia (*POU2/3, six-1/2-2*), muscle (*myoD, snail*), CNS (*pax6A*), eye (*ovo*), pharynx (*foxA*); see also Supplementary Table 3. Each column depicts different identified SpNBs and each row depicts additional identified examples. Representative images are from *n* = 10 animals per comparison. Scale bars, 5 μm. **b** SpNB are interspersed by non-self neoblasts in the planarian tail. Colored surfaces represent detected neoblasts (yellow) and non-identified neoblasts (cyan). Numbers represent the proportion of a specialized class compared to all detected neoblasts. Scale bars, 50 μm. **c** Surface-to-surface distance measurements of specialized neoblasts to their closest self or non-self neoblast from regions identified in

**b** and Supplementary Fig. 16. The bounds of box correspond to 25th and 75th percentiles, whiskers represent 75th percentile +1.5*interquartile range (IQR) and 25th percentile −1.5*IQR. Minimal and maximal points are represented by whisker bounds or points if <25th percentile −1.5*IQR or >75th percentile +1.5*IQR respectively. Two-sided Welch's *t*-test, *$p$ = 0.0465, ****$p$ < 0.0001, n.s. not significant. **d** Nearest neighbor (NN) identities of identified specialized neoblasts for self-class (yellow) or non-self-class (cyan) neoblasts identified in **b** and Supplementary Fig. 16. N values for **c** and **d** =139 (Epidermis), 42 (Intestine), 21 (Muscle), 28 (Protonephridia). Epi epidermal-, Int intestinal-, Mus body-wall muscle-, and Nep protonephridia-specialized neoblasts. Source data are provided as a Source Data file.

neoblast fate pair tested. If the neoblast microenvironment influences its fate, spatial neighborhoods of neoblasts enriched for particular fates would be expected, although clusters need not be exclusive for a particular fate. To determine the identity of cells in local neoblast neighborhoods, we generated 3-dimensional Voronoi diagrams using neoblast centroid coordinates. Voronoi analysis generates space-filling polygons in three dimensions around each coordinate in a given set; the topological dimensions of the polygons represent the relative distances between nearby coordinates (Supplementary Fig. 22a). We defined neighbors as the immediate first-degree contacts of a Voronoi polygon representing a specialized neoblast of interest. The Voronoi polygons did not necessarily approximate cell boundaries, as intervening non-neoblast cells were not included. We assessed neighborhood composition for all investigated specialized neoblasts in pairwise comparisons to determine if class enrichment was present (Fig. 3d, Supplementary Fig. 22b, c, Supplementary Table 4). Specialized neoblasts displayed similar frequency distributions of self and non-self neoblasts in their immediate neighborhoods, often demonstrating bias in favor of non-self-classes, especially when the non-self-class was more abundant. For example, 44% of muscle-specialized neoblasts had no muscle-specialized neoblasts in their immediate neighborhood, whereas 34% of the time one third or more of their neighbors were epidermal (Fig. 3d). No muscle-specialized neoblast had more than 50% of its neighborhood comprised of muscle-specialized neoblasts (Fig. 3d). Specialized neoblasts from more abundant classes (such as

epidermal specialized neoblasts) had a higher proportion of same-class neighbors compared to less abundant specialized neoblast classes, as expected (Fig. 3d, Supplementary Fig. 22b, c). For example, epidermal specialized neoblasts are ~30% of all neoblasts whereas muscle-specialized neoblasts are ~5%; random sampling should therefore result in more epidermal specialized neoblasts in a given neighborhood than the less abundant muscle-specialized neoblasts[20,28]. Observed nearest neighbor measures were compared to randomized simulations using neoblast position data and overall frequencies. Many cases were similar to random; there were some cases where the nearest neighbor identities of tested neoblast classes were biased toward the self-class as compared to random simulations, but even in these cases their immediate spatial neighborhoods exhibited extensive heterogeneity (Fig. 3e, Supplementary Figs. 22 and 23).

These trends were present regardless of which neoblast class pairs were queried, with neighborhood class distribution biases more closely tied to the gross abundance of the neoblast class than to the query class identity. The observed deviations from random nearest neighbor frequency could reflect influence of position on fate, as described above, and/or moderate bias for fate to resemble that of neoblasts related by division.

Eye and pharyngeal fates are regionally specified in broad regions, but near their target tissues (Fig. 3f, g, Supplementary Fig. 24a, b). Consistent with other tested fates, eye and pharyngeal specialized

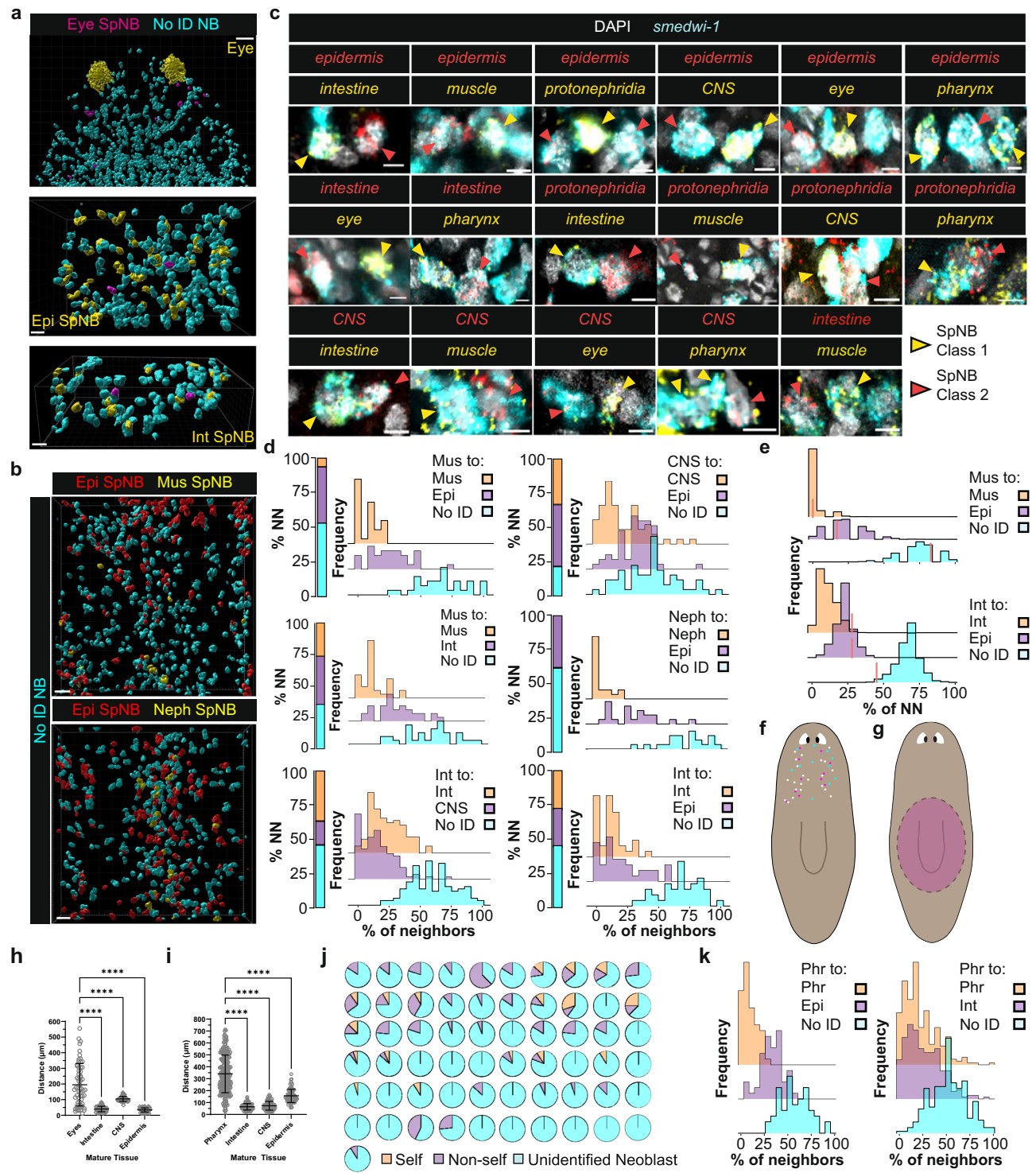

neoblasts were on average closer to different mature tissues than to the eye or pharynx respectively (Fig. 3h, i). Experiments applying lethal doses of radiation (which ablates all neoblasts and halts new cell production) at an intermediate regeneration timepoint resulted in eye growth by approximately the number of cells present within eye progenitor tails, suggesting that progeny of distant eye-specialized neoblasts incorporate into the eye in regenerating heads[15]. Furthermore, neoblast neighborhoods of eye-specialized neoblasts were heterogenous; in many cases, no same-class neoblasts were present in the immediate vicinity (Fig. 3j, Supplementary Fig. 24c). Neoblast neighborhoods around pharyngeal specialized neoblasts were also highly heterogenous (Fig. 3k, Supplementary Fig. 24d).

Using MERFISH, the neighborhoods of specialized neoblasts were directly visualized to be highly heterogeneous and intermingled (Fig. 4a, Supplementary Fig. 25). The heterogenous fate-specification pattern spanned body locations, indicating fate choice intermingling is a general property throughout the neoblast population. Overall, neoblast neighborhood data indicate a striking degree of fate choice spatial heterogeneity, supporting a model in which fate specification occurs in a highly intermingled "salt-and-pepper" distribution.

## Intermingled fate specification in neoblast colonies
We next assessed fate specification in small neoblast colonies after subtotal irradiation, in which most but not all neoblasts are cleared

**Fig. 3 | Fate specification is highly intermingled in planarian stem cells. a** (Top) eye-specialized neoblasts interspersed among non-self neoblasts, which frequently adopt epidermal (middle) or intestinal (bottom) fates. SpNB, specialized neoblast. No ID NB, neoblasts not labeled with assessed FSTFs. Scale bars, (top) 30 μm, (middle and bottom) 10 μm. Representative images from $n = 3$ animals. **b** Muscle and protonephridia fates in prepharyngeal regions can be intermingled with neoblasts choosing epidermal fate. Scale bars, 20 μm. Representative images from $n = 5$ animals. **c** Directly neighboring neoblasts can make different fate decisions. Scale bars, 5 μm. Representative images from $n = 10$ animals per comparison. Genes for tissue-specific FSTF pools: epidermis (*soxP-3*), intestine (*hnf-4, gata4/5/6-1*), protonephridia (*POU2/3, six-1/2-2*), muscle (*myoD, snail*), CNS (*pax6A*), eye (*ovo*), pharynx (*foxA*); see also Supplementary Table 3. **d** Neighborhood composition analysis of specialized neoblast classes. Stacked bar plots represent percentage of nearest neighbors for sampled cells of a given specialized neoblast query class that are of labeled identities. Ridgeline plots represent percentage of cells in the Voronoi tessellation neighborhoods of sampled cells of a given specialized neoblast query class that are of labeled identities. Top left, middle left, and top right are neoblasts from the pre-pharyngeal region of animals. All other plots are neoblasts from animal tail regions. **e** Randomized nearest neighbor identity composition for a

given query class ($n = 1000$ simulations) compared to observed nearest neighbor (NN) percentage (red bar). All plots are neoblasts from tail regions. **f** Cartoon depicting the specification region of eye-specialized neoblasts in homeostasis. Circles represent individual mapped eye-specialized neoblasts and colors represent biological replicates. Data collected from 3 independent animals. **g** Cartoon depicting the specification region of pharyngeal specialized neoblasts. **h** Distance measurements of eye-specialized neoblasts to eyes, intestine, CNS, or epidermis. $N = 61$ from 4 independent animals. **i** Distance measurements of pharyngeal specialized neoblasts to the pharynx, intestine, CNS, or epidermis. $N = 249$ from 3 independent animals. **j** Voronoi neighborhood identity composition of eye-specialized neoblasts. $n = 41$ pooled from 4 independent animals. Individual neighborhood ridgeline plots available in Supplementary Fig. 24c. **k** Ridgeline plots of Voronoi neighborhood identity composition for pharyngeal specialized neoblasts. N values for **d**, **e**, and **k** provided in Supplementary Table 4. Two-way ANOVA with Dunnett's test in **h** and **i**, ****$p < 0.0001$. The central line represents the mean and error bars represent standard deviation in **h** and **i**. Mus muscle, Epi epidermal, CNS central nervous system, Int intestine, Neph protonephridia, Phr pharyngeal. Source data are provided as a Source Data file.

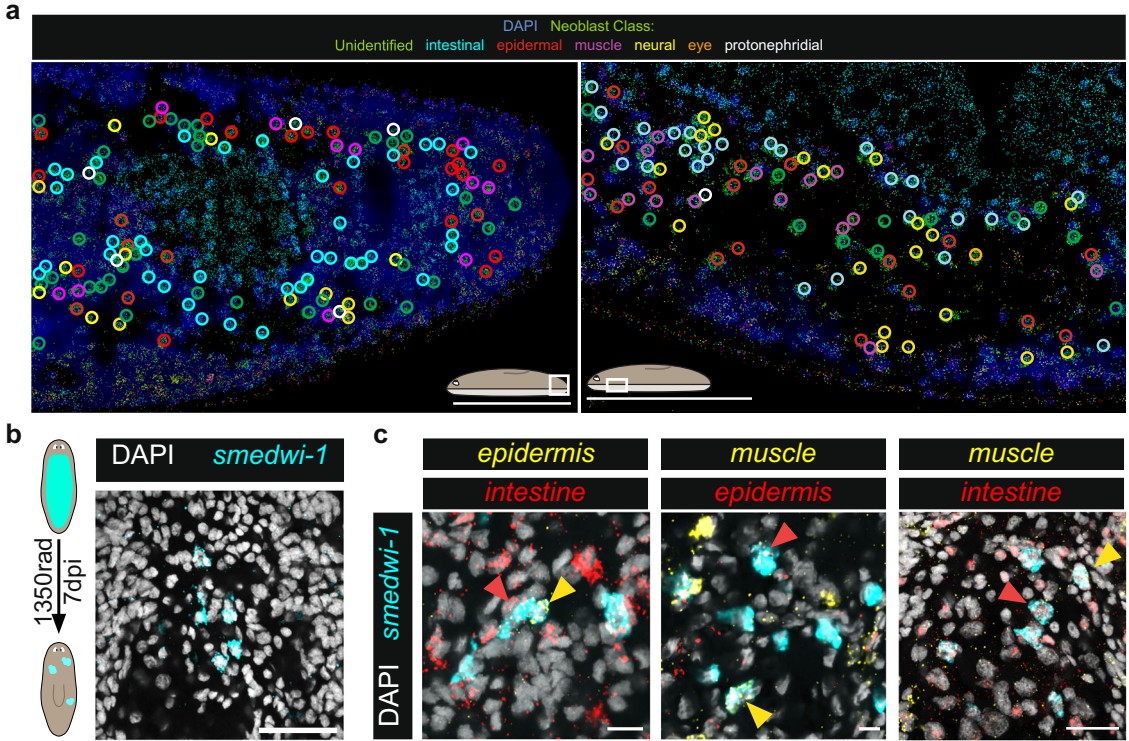

**Fig. 4 | Fate specification is highly intermingled. a** Highly intermingled specialized neoblasts can be observed in the planarian tail (left) and pre-pharyngeal region of the animal (right). Right image corresponds to the same inset region of Fig. 1a. Colored circles represent identified specialized neoblasts. Scale bars, 100 μm. Representative images from $n = 3$ animals. **b** Generation of small neoblast colonies by subtotal irradiation. Scale bar, 25 μm. Representative image from $n = 20$ animals. **c** Different fate choices in early neoblast colonies after subtotal irradiation.

Scale bars, (left) 15 μm, (middle) 10 μm, (right) 25 μm. Identified neoblast colonies were located in ventral regions of animals. Arrows depict identified specialized neoblasts. Representative images from at least $n = 3$ identified double-positive colonies. Genes for tissue-specific FSTF pools: epidermis (*soxP-3*), intestine (*hnf-4, gata4/5/6-1*), muscle (*myoD, snail*); see also Supplementary Table 3. dpi days post irradiation.

from the animal (Fig. 4b, Supplementary Fig. 26a). Surviving neoblasts can clonally expand to generate neoblast colonies related by lineage. In these colonies, fate choices are newly made during proliferation from a single progenitor. Previous work demonstrated that colony composition of epidermal, pharyngeal, muscle, intestinal, and neural specialized neoblasts individually did not substantially deviate from random, given the overall homeostatic frequency of these classes[19]. We observed that colonies contained multiple distinct specialized neoblast classes, demonstrating specification of different fates in close proximity and from neoblasts closely related by lineage (Fig. 4c,

Supplementary Fig. 26b). For example, colonies with one epidermal and one intestinal specialized neoblast, as well as colonies with one muscle and one epidermal specialized neoblast were observed (Fig. 4c). These data provide independent support to the interpretation that specification of different fates can occur in close proximity and in close lineage relatives.

## Fate specification is intermingled in regeneration
Regeneration requires the production of new fates at wounds to restore missing tissues[29]. After major injury, neoblasts exhibit

increased proliferation at wounds and form a blastema, a regenerative bud where differentiation of new tissues occurs[30–32]. Depending on injury type, multiple fates might need to be specified within a small region of neoblasts at a wound site[29]. We performed pair-wise FISH of different specialized neoblast classes in 72 h post amputation (hpa) regenerating animals and observed intermingled fate choices across these dense neoblast classes involved in blastema formation (Fig. 5a, Supplementary Fig. 27a–f). Intermingled specialized neoblast distributions were also directly visualized by MERFISH at anterior- (72hpa) and posterior- (96hpa) facing wounds (Fig. 5b, c, Supplementary Fig. 28).

We assessed neoblast neighborhoods at anterior-facing wounds to determine if fate specification occurred in spatially distinct clusters (Fig. 5d, e, Supplementary Fig. 29). Consistent with homeostatic conditions, neoblast neighborhoods were found to be heterogenous with frequent non-self neoblast neighbors (Fig. 5d, e, Supplementary Fig. 29a). Regionally specified eye-specialized neoblasts are found only at anterior-facing wounds (Supplementary Fig. 29b, c)[21]. Neoblast neighborhoods around eye-specialized neoblasts were highly heterogenous, with many eye-specialized neoblasts possessing no immediate eye-specialized neoblast neighbors (Fig. 5e, Supplementary Fig. 29d). Additionally, many specialized neoblasts at wound sites were observed to be a similar distance or closer on average to different mature tissues than to their corresponding tissue, particularly parenchymal cell types (Supplementary Fig. 30). In sagittal blastemas, specialized neoblasts of different fates were spatially intermingled and neural specialized neoblasts were distant on average from target regenerating nerve cords, positioned at similar distances to different mature tissues and significantly closer to parenchymal cell types (Fig. 5f, g, Supplementary Fig. 30d). Overall, these data indicate that specification of distinct fates can occur in close proximity of one another in regeneration.

### Migratory targeting of post-mitotic progeny generates anatomical pattern

If fate choices in neoblasts are made in a spatially heterogenous manner, migratory assortment of neoblast progeny might have a much more central role in pattern formation than the pattern of fate choice itself. Several lines of evidence support this hypothesis. Whereas neoblasts are present throughout the animal in the parenchymal space, they are excluded from several regions of the animal including the pharynx and head tip anterior to the eyes. BrdU labeling specifically labels neoblasts and is retained in their post-mitotic descendant cells. BrdU labeling demonstrates renewal of the head tip that lacks neoblasts, indicating movement of post-mitotic neoblast descendants into this region[11]. Similarly, pharyngeal progenitors are specified from neoblasts outside of the pharynx, and post-mitotic neoblast descendants migrate into the organ, which itself lacks any dividing cells[28,33]. Epidermal progenitors transit maturation stages as cells migrate through body-wall muscle into the epidermis[20,34,35]. Partial irradiation and transplantation experiments indicate that neoblasts do not rapidly migrate away from their original location, but that post-mitotic epidermal progenitors, the descendants of epidermal specialized neoblasts, can[12,25,26,36,37].

To further assess the ability of neoblast progeny to migrate and target distant tissues, we partially irradiated planarians with X-rays by shielding a stripe in the trunk region with lead, followed two days later by an EdU pulse to label remaining neoblasts (Fig. 6a–c, Supplementary Fig. 31a). EdU+ differentiated cells observed thereafter must have been specified within the remaining neoblast zone. Surviving neoblasts in the shielded region exhibited intermingled fate specification, with many instances of directly adjacent epidermal and intestinal specialized neoblasts present (Fig. 6b, Supplementary Fig. 31b). We observed robust incorporation of EdU+, smedwi-1– neoblast progeny into the cephalic ganglia and the epidermis distant from the remaining region occupied by neoblasts (Fig. 6c, Supplementary Fig. 31c, d).

To assess migration in non-irradiated contexts, we first identified smedwi-1-low/FSTF+ cells, which correspond to early post-mitotic progenitors and measured their distances to target tissues[19]. These post-mitotic progenitors were closer to target tissues than corresponding specialized neoblasts, consistent with progenitor movement towards their target (Supplementary Fig. 32). We pulsed uninjured animals with EdU to label neoblasts, after which animals were fixed and assessed seven days later. This allows for visualization of post-mitotic neoblast descendant localization. We observed EdU+ cells in the head tip, including in anterior-most cintillo+ sensory neurons and collagen+ muscle cells, beyond the regions containing neoblasts, indicating neoblast descendant cells moved to these distal locations (Fig. 6d, e, Supplementary Fig. 33a–c). Additionally, we pulsed animals with EdU 3 days post decapitation, and observed EdU+ neoblast descendant cells in the anterior end of newly forming neural tissues 6 dpa, beyond the region containing neoblasts (Fig. 6f, Supplementary Fig. 33d). Taken together, these data indicate that post-mitotic neoblast progeny assort from an intermingled field of specialized neoblasts by migrating to distant locations for maintenance and formation of tissue pattern.

## Discussion

The ability of progenitors to correctly choose their fate among a multitude of possibilities is a critical challenge in regeneration. We applied MERFISH in planarian tissues to spatially characterize multiple specialized neoblast classes simultaneously in sections. In combination with systematic three-dimensional analyses of specialized neoblasts, we demonstrate that fate specification in planarian stem cells is highly intermingled and distributed in spatially coarse domains. We suggest a model for planarian tissue maintenance and regeneration with two key components. First, pattern formation and maintenance are driven primarily by the migratory sorting of highly intermingled progenitors rather than by the spatial pattern of stem cell fate choice (Fig. 7a). Second, spatial position biases neoblast fate choices available in broad regions, but cell-internal processes drive fate choice in locally highly intermingled neighborhoods. We suggest that positional information in the form of position control genes influences the options available and choice probabilities for a stem cell, but that the choice among options is largely up to the neoblast itself (Fig. 7b)[29,38].

It remains possible that an untested planarian progenitor class might display highly structured organization of fate choice. Similarly, local processes might also exist that influence the intermingled nature of fate choice, such as communication between neighboring neoblasts or differentiated cells. However, because no overt bias towards or away from particular neoblast classes was seen in neighborhood composition or nearest neighbor analyses, such a hypothetical process does not appear to be the fundamental mechanism underlying the pattern observed. For example, nearest neighbor analyses showed that neighbors could readily choose self or non-self, and that the non-self choice was not always of a particular type. It is also possible that differentiated tissues could influence final fates for certain neoblast descendants. However, transplant of eyes outside of the normal specification zone of eye progenitors showed no evidence of influence on nearby progenitors to maintain the ectopic eye[39]. Furthermore, no evidence of hybrid-fate cells is observed by single-cell RNA sequencing, which might be expected under this fate-switching model[13].

The proposed model also does not address the potential presence or absence of a fully uncommitted neoblast population[40]. Single-cell transplantation experiments showed that at least some TSPAN-1+/tgs-1+ neoblasts can repopulate the neoblast compartment[40]. However, tgs-1+ neoblasts frequently display neural specialization and further single-neoblast transplantation experiments into lethally irradiated hosts demonstrated a 2-fold higher colony-formation rate than would be expected if only fully uncommitted neoblasts were capable of colony formation, suggesting that at least some specialized neoblasts can be clonogenic[19]. Similarly, no single tested neoblast class was present in

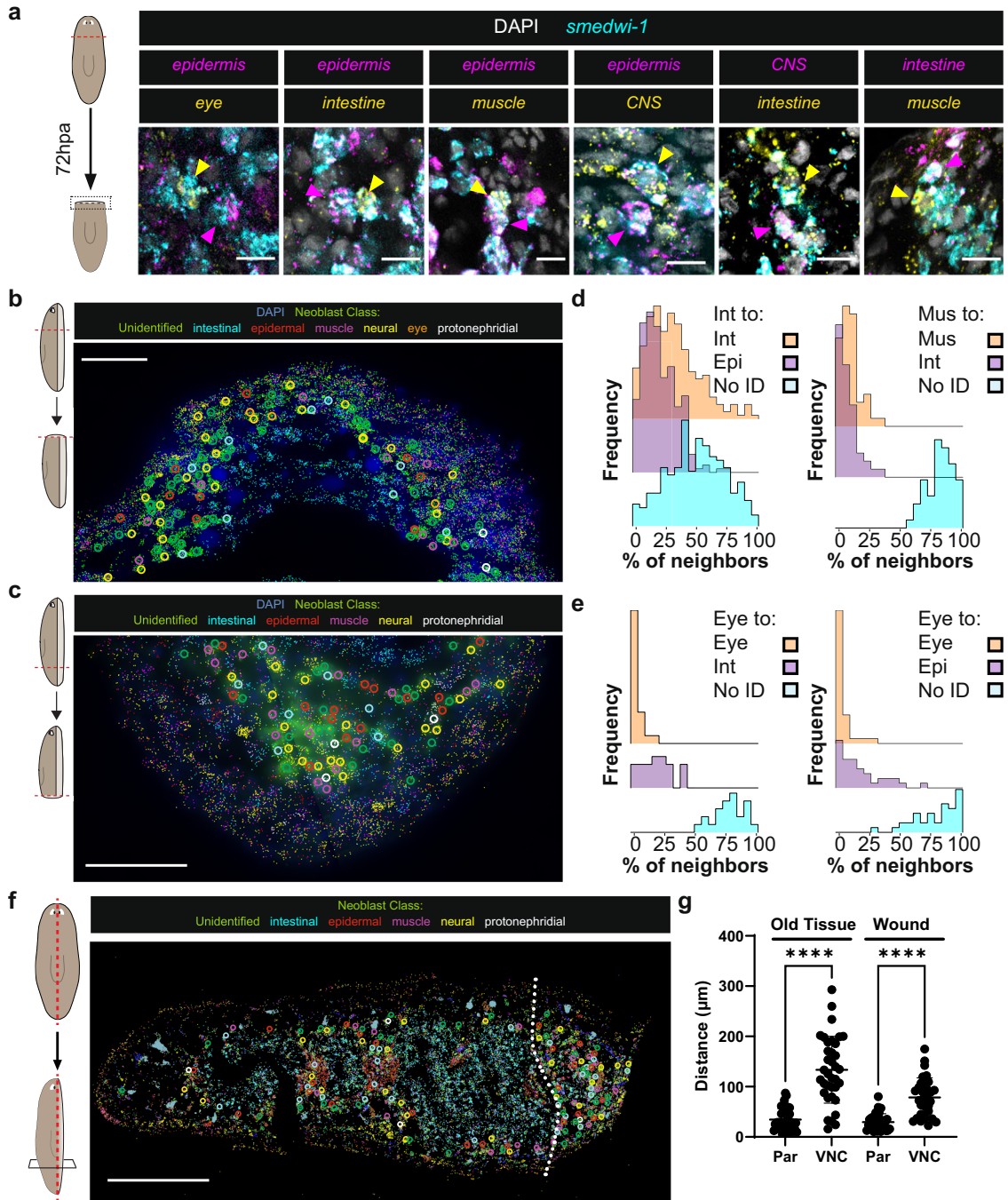

**Fig. 5 | Fate specification is intermingled in regeneration. a** Different fates can be specified in neighboring neoblasts at wound sites in regeneration. Pairs identified at anterior-facing wounds 72 hours post amputation. Scale bars, 10 μm. Representative images from *n* = 10 animals. Additional examples available in Supplementary Fig. 27. Arrows depict identified specialized neoblasts. Genes for tissue-specific FSTF pools: epidermis (*soxP-3*), intestine (*hnf-4, gata4/5/6-1*), muscle (*myoD, snail*), CNS (*pax6A*), eye (*ovo*); see also Supplementary Table 3. **b** Intermingled specialized neoblasts of distinct fate can be observed in an anterior-facing blastema 72hpa. Scale bar, 50 μm. **c** Intermingled specialized neoblasts can be observed in a posterior-facing blastema 96hpa. Scale bar, 50 μm. Representative images from *n* = 4 animals in **b** and **c**. Additional examples available in Supplementary Fig. 28. **d** Neighborhood analysis of intestinal and muscle-specialized neoblasts at anterior-facing wound sites 72hpa. N values provided in Supplementary Table 4.

**e** Neighborhood analysis of eye-specialized neoblasts at anterior-facing wound sites 72hpa. N values provided in Supplementary Table 4. **f** Specialized neoblasts identified in in a transverse section of an animal 5 days post sagittal amputation. Dotted line represents boundary between old pre-existing tissue (left) and regenerating tissue (right). Scale bar, 100 μm. Representative image from *n* = 3 animals. **g** Distance measurements of ventral neural specialized neoblasts to ventral nerve cords or parenchymal cell types in old and regenerating tissue. *N* = 36 in old (left) and 31 in regenerating (right) tissues. Two-sided Welch's *t*-test, ****p* < 0.0001. The central line represents the mean and error bars represent standard deviation. Colored circles in **b**, **c**, and **f** depict identified specialized neoblasts. Int intestinal, Epi epidermal, Mus muscle, VNC ventral nerve cord, Par parenchymal cell types. Source data are provided as a Source Data file.

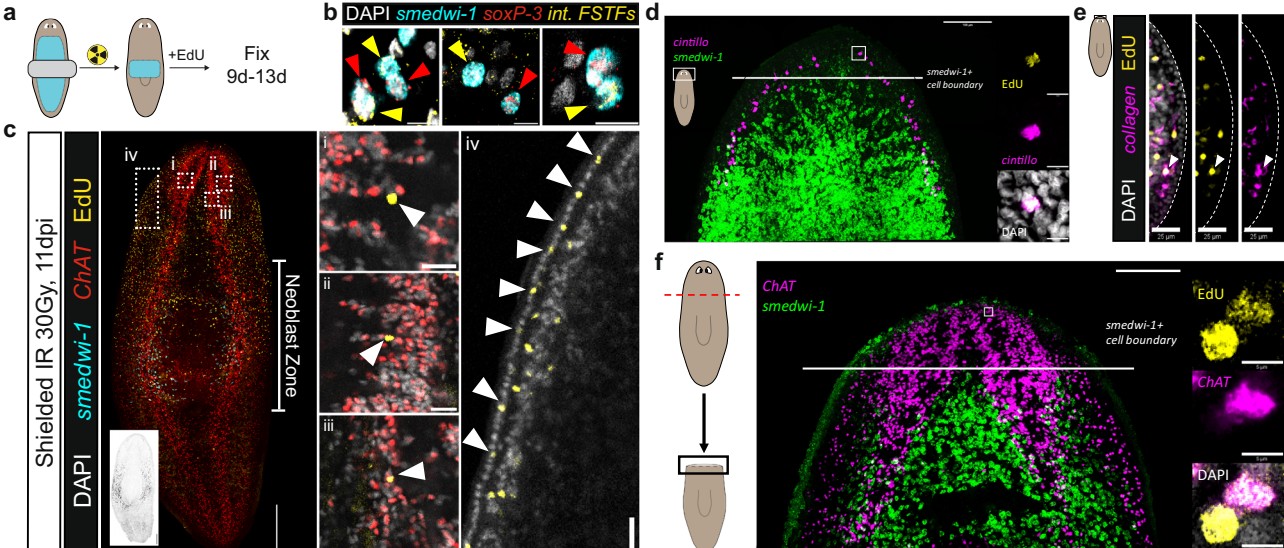

**Fig. 6 | Progenitors can migrate and be targeted to mature tissues. a** Lead shielding preserves a region of neoblasts (cyan) during lethal irradiation. Animals were pulsed with EdU after neoblast clearing and fixed 9-13dpi. **b** Specialized neoblasts are intermingled in the shielded region after irradiation. Red arrows, epidermal specialized neoblasts; yellow arrows, intestinal specialized neoblasts. Scale bar, 5 μm. Representative images from *n* = 10 animals. Arrows depict identified specialized neoblasts. Int FSTFs represent a probe pool containing *gata4/5/6-1* and *hnf-4*. **c** Incorporation of new progeny (EdU⁺ cells) into tissues far from the region containing surviving neoblasts (neoblast zone) can be observed in the (i-iii) cephalic ganglia and the (iv) epidermis. Inset depicts inverted and saturated *smedwi-1* fluorescence signal. Scale bars, 250 μm. White arrows indicate newly incorporated EdU⁺ cells. Representative images from *n* = 20 animals.

**d** Incorporation of new *cintillo*⁺ sensory neurons (EdU⁺) at the anterior head tip of the animal. Inset region (right) depicts EdU⁺ *cintillo*⁺ cell. Scale bars, 100 μm (left), 5 μm for insets. *smedwi-1*+ cell boundary represents the anterior-most boundary at which a neoblast was detected. Representative images from *n* = 10 animals. **e** Incorporation of a new *collagen*⁺/EdU⁺ muscle cell at the anterior head tip of the animal, indicated by the white arrow. Scale bars, 25 μm. Representative images from *n* = 10 animals. **f** Animals were decapitated below the auricles to completely remove the cephalic ganglia, pulsed with EdU at 4 dpa, and assessed at 6 dpa for new cell incorporation (left). Inset region (right) depicts an identified *ChaT*⁺/EdU⁺ cell at the anterior of the newly regenerating brain. *smedwi-1*⁺ boundary line represents the anterior-most boundary at which a neoblast was detected. Scale bars, 100 μm (left), inset regions, 5 μm. Representative images from *n* = 10 animals.

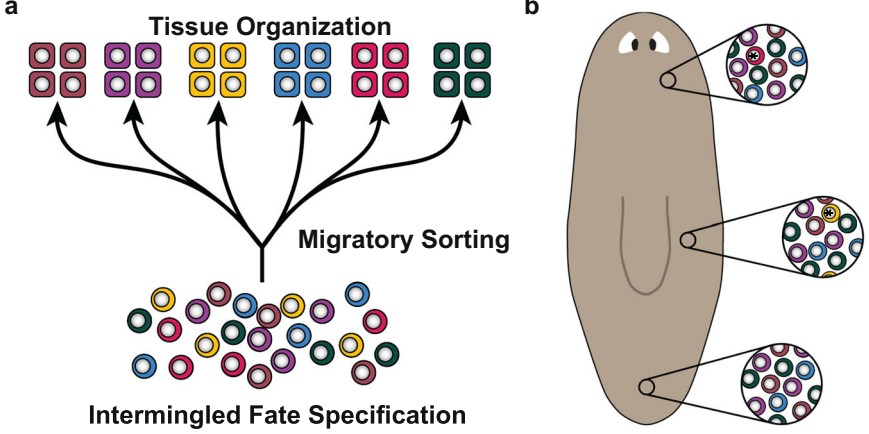

**Fig. 7 | A model for the intermingled specification of fate in planarian neoblasts. a** A model for the formation of organized anatomy patterning from spatially intermingled fate specification in neoblasts. Fate choice occurs in neoblasts in a highly intermingled manner, with different fates able to be specified in adjacent neoblasts. Post-mitotic progenitors then migrate and incorporate into target

tissues, generating anatomical organization. **b** A model depicting regional influence on neoblast fate choice across the anteroposterior axis. The * symbol in the nucleus indicates a regionally specified specialized neoblast (top and middle insets). The dorsoventral axis (not shown) is also known to influence epidermal fate[38].

all early neoblast colonies after subtotal irradiation[19]. Asymmetric neoblast division outcomes and lower expression of FSTFs in G1 neoblasts were also observed, supporting a non-hierarchical neoblast lineage model called the single-step fate model[19]. In this model, specialized neoblasts do not typically undergo multiple divisions with fate retained, as would be the case for lineage-amplifying or gradual fate-restricting divisions. Further study of pluripotency and fate decisions in neoblasts presents an important area for future investigation. The

highly intermingled nature of neoblast classes is consistent with the single-step fate model, because if fate was retained through division, closely related neoblasts by division should be physically closer than distantly related neoblasts. However, it remains possible that relationship by lineage has a moderate biasing effect on fate choice[19].

Drosophila neuroblasts can display distinct fates over divisions in a stereotypical order, called temporal patterning[41]. It is possible that individual neoblasts could similarly adopt fates in a prescribed order

over divisions. Prior work, however, revealed a high degree of fate-composition heterogeneity in early 4-cell neoblast colonies after sub-total irradiation, in which colonies can consist of 1–4 positively identified specialized neoblasts of same class, with frequencies of these classes consistent with random sampling[19]. This phenomenon was observed for epidermal, intestinal, muscle, and pharyngeal specialized neoblasts[19]. A stereotyped order of fate transitions could result in fate distributions in colonies that deviate from predictions with random sampling. Spatial analyses here also did not reveal overt pattern in nearest neighbors (all possible nearest neighbors occurred for all pairwise tests made) or with neighborhoods, although intermixing of cells in different stages of a stereotyped order could obscure pattern. Whereas a prescribed fate-choice order has thus not been revealed, it remains possible that the specification of particular fates could occur in some complex order, which will be an important target for future studies. It is also possible that only, or preferentially, specification events within some permissible zone or distance from a target result in productive outcomes. Neoblasts and their progeny do not contribute significantly to overall homeostatic cell death levels, however, which would be expected if distal specialized neoblasts were pruned[42]. Furthermore, if distant specialized neoblasts were pruned, or switched fates, more structure would be expected in neoblast neighborhoods and distances to target tissues than was observed. Shielded irradiation experiments also demonstrate that post-mitotic progenitors can be targeted to distant tissues (Fig. 6c). Migratory targeting was also observed in non-irradiated homeostatic and regeneration conditions (Fig. 6d–f, Supplementary Figs. 32 and 33). Similarly, migration and targeting from distant specification events has been demonstrated for the regenerating eye, homeostatic epidermis, head tip, and pharynx[11,15,21,33–35,43,44]. Finally, inhibition of migration (through *β-integrin-1* RNAi) results in aberrant neural ectospheres associated with the inability for progenitors to be targeted to their target tissues[45,46]. These findings together indicate that distal fate specification events are often followed by migratory sorting of post-mitotic progenitors for maintenance and regeneration of tissue pattern.

Stem-cell fate specification in a distributed pattern yields benefits for regeneration that might underlie emergence of this mechanism. For instance, fate-specified progenitors can remain present after certain injuries removing their target tissue because of their dispersed nature, allowing regeneration to initiate rapidly with existing progenitors, especially in the case of local injuries. To be broadly distributed, but not over abundant, specification of a particular fate might need to be scattered across a broad stem-cell field. Many cell fates spread over broad distances in this way is compatible with an intermingled fate-specification process. Furthermore, if fate specification was heavily influenced by a local target tissue itself, such influence could be lost with local injury. Similarly, if fate specification was primarily regulated by positional information in very local pattern at the target tissue location, then local injury could remove such information. Restoring the missing environment without the necessary information to specify the identity of the cells that occupy it would present a major challenge, and specifying progenitors in a broad domain presents one solution to this challenge. Finally, positional information mechanisms accessible to adult tissues might not readily interface with fate-specification mechanisms in a manner that could carry enough information to precisely generate the large array of choices needed and in the correct proportions. Multipotent progenitors have been identified in other animals capable of whole-body regeneration including other platyhelminthes, acoels, urochordates, and cnidarians[27]. Further work in dissecting the spatial patterning of fate choice in these additional models will provide deeper insight into the fundamental mechanisms driving whole-body regeneration.

Stochastic state choice occurs in multiple contexts of development and biology. For example, *Drosophila* photoreceptor fate is stochastically established through random expression of the transcription factor Spineless[47,48]. Selection of mouse olfactory receptor gene expression involves regional influence and stochastic choice[49,50]. In vertebrates, bi-potent progenitors contributing to both ectodermal and mesodermal fates post-gastrulation exist in posterior elongation, termed neuromesodermal progenitors (NMPs)[51,52]. It has been proposed that NMPs exhibit a high degree of gene expression heterogeneity and undergo a period of stochasticity prior to convergence to either neural or mesodermal fate[53]. Spatially coarse fate specification can be followed by pattern refinement in additional contexts as well. In zebrafish neural tube development, intermingled progenitor fate specification is influenced by Shh signaling and is followed by assortment into pattern[54]. Similarly, an initial mosaic pattern of different motor neuron classes can be sorted to distinct pools in the lateral motor column of the vertebrate spinal cord in a cadherin-dependent manner[55]. We suggest that coarse and intermingled fate specification followed by progenitor sorting might prove to be a widespread solution to tissue pattern formation challenges, particularly in the context of regeneration.

## Methods

### Animal husbandry and surgical procedures

*Schmidtea mediterranea* clonal asexual strain CIW4 animals were maintained in 1x Montjuic water (1.6 mmol/l NaCl, 1.0 mmol/l CaCl2, 1.0 mmol/l MgSO4, 0.1 mmol/l MgCl2, 0.1 mmol/l KCl and 1.2 mmol/l NaHCO3 prepared in Milli-Q water) at 20 °C in the dark. Animals were fed blended calf liver 1-2 times a week and washed twice per week. Animals were starved 7 days prior to use for all experiments.

For regeneration experiments, animals were amputated using the following methodology. Animals were placed on moistened filter paper on a cold block to minimize movement and cut with a clean scalpel. Resulting fragments were then placed into and maintained in 1x Montjuic water with 0.1% gentamicin at 20 °C in the dark to recover prior to fixation. For generation of anterior-facing wounds, animals were decapitated using the auricles as a landmark and fixed 72 h post amputation (hpa). For generation of posterior-facing wounds, animals were cut at the midpoint between the mouth of the pharynx and posterior edge of the tail and fixed 96hpa. For sagittal wounds, animals were cut along the midline and fixed 5 days post amputation. All samples were immediately utilized for FISH and MERFISH experiments.

### Fluorescence in situ hybridization (FISH)

Planarians were sacrificed in a 5% N-acetyl-cysteine/1x phosphate buffered saline (PBS) solution and fixed in a 4% fresh paraformaldehyde/1x PBS (0.1% Triton-X100) solution. Samples were stored in methanol at −20 °C until use. Animals were bleached in 1x SSC/5% de-ionized formamide/1.2% hydrogen peroxide for 2 h on a light table. Animals were permeabilized in a 2 mg/mL Proteinase-K/0.1% sodium dodecyl sulfate/1x PBS (0.1% Triton-X100) solution and hybridized overnight with RNA probes diluted 1:800 in hybridization buffer [50% deionized formamide, 5x SSC, 1 mg/mL yeast RNA, 1% Tween-20, 5% dextran sulfate] at 56 °C. Animals were blocked prior for 1–2 h prior to antibody treatment overnight with anti-DIG-POD (1:1000, Roche; blocking solution of 10% Western blocking reagent (Roche)), anti-FITC-POD (1:2000, Roche; blocking solution of 5% Western blocking reagent (Roche) and 5% 10x Casein solution (Sigma Aldrich)), or anti-DNP-HRP (1:100, Perkin-Elmer; blocking solution of 5% 10x Casein solution (Sigma Aldrich) and 5% heat inactivated horse serum) at 4 °C. For tyramide signal amplification, animals were washed in borate buffer (0.1 M boric acid, 2 M NaCl, pH 8.5) and treated for 10 minutes in TSA buffer (borate buffer with 1% 20 mg/mL 4-IPBA, 0.0003% hydrogen peroxide) containing rhodamine (1:1000), fluorescein (1:1500), or Cy5 (1:300) tyramide. Prior to additional color development, animals were treated with 1% sodium azide for 1.5 h at ambient temperature to

inactivate peroxidase activity. Prior to mounting on glass slides, animals were placed in a 1 mg/mL DAPI solution overnight at 4 °C.

## Cryo-sectioning

Planarians were fixed as described above. Samples were stored in methanol at −20 °C until use. Samples were rehydrated in 1x PBS. Samples were transferred to fresh OCT (Tissue-tek) within a 10 mm × 10 mm × 5 mm cryomold (Tissue-tek) and arranged prior to being flash frozen in 2-methylbutane (Honeywell) immersed in a liquid nitrogen bath. Tissue blocks were then acclimated to −20 °C in a cryostat (Leica CM3050 S) for at least 30 minutes prior to sectioning at 10 μm thickness. Sections were collected on glass coverslips and utilized immediately in FISH protocols, or collected on MERSCOPE slides (Vizgen) and utilized immediately for MERFISH experiments.

## F-ara-EdU treatment

Animals were soaked in a 1.25 mg/mL F-ara-EdU (Click Chemistry Tools)/1x Monjuic water solution for 8 h for use in short-pulse experiments and 18 h for shielded irradiation experiments and for incorporation experiments in homeostasis and regeneration. Relevant chase periods are specified in corresponding figure legends. Prior to fixation, animals were washed and stored in a solution of 10x Instant Ocean (Instant Ocean). EdU was developed using TAMRA-azide (Sigma Aldrich) prior to tyramide signal amplification for FISH.

## Sublethal irradiation

Animals were irradiated using a dual Gammacell-40 137-cesium source to deliver 13.5 Gray. Animals were kept in 1x Montjuic water with 0.1% gentamicin (Gibco).

## Shielded irradiation

Animals were anesthetized in a 0.2% chlorotone (1,1,1-tricloro-2-methyl-2-propanol, Sigma Aldrich)/1x Montjuic water for 4 min prior to arrangement onto wet filter paper (GE Healthcare) in a plate on an ice bath. A lead shield was then placed above the arranged animals prior to placement of the plate and ice bath into the irradiation chamber (X-Rad320, Precision X-Ray Irradiation). Samples received a total of 30 Gray of unidirectional irradiation. Animals were then gently washed and kept in 1x Montjuic water with 0.1% gentimicin (Gibco).

## MERFISH library design and synthesis

Genes were selected based on known roles as FSTFs and enrichment in differentiated tissues including neoblasts, muscle, intestine, epidermis, protonephridia, neurons, *cathepsin*+ cells, parenchymal cells, and pharynx. Libraries were synthesized by Vizgen (Cambridge, MA) for use on the MERSCOPE platform.

## MERFISH sample preparation and imaging

Planarians were fixed and sectioned as described above. Sections were prepared according to MERSCOPE sample preparation guidelines (Vizgen, Cambridge, MA). Briefly, sections were washed in 0.1 M phosphate buffered saline (PBS, pH 7.4) and placed into 70% ethanol (v/v) solution overnight at 4 °C to permeabilize tissue. Sections were then hybridized with an encoding probe mix (Vizgen) for 36–42 h in a humidified incubator at 37 °C. Samples were then embedded in a gel embedding solution [gel embedding premix (Vizgen), 5 mL; 10% ammonium persulfate solution (Thermo Fisher), 25 microliters; N,N,N′,N′-tetramethylethylenediamine (Sigma Aldrich), 2.5 microliters]. Tissues were cleared in clearing solution [clearing premix (Vizgen), 5 mL; proteinase-K (New England Biolabs), 50 microliters] for at least 24 h in a humidified 37 °C incubator. Tissues were then incubated with 4′,6′-diamidino-2-phenylindole (DAPI) and polythymine (polyT) staining reagent (Vizgen) and imaged using MERSCOPE (Vizgen). Images were processed using MERSCOPE Visualizer software (Vizgen).

## FISH image acquisition and processing

Fluorescence images were acquired using Leica SP8 and Leica STELLARIS 5 confocal microscopes. ImageJ software (Fiji) and QuPATH was used for the processing and quantification for single z-plane images. QuPATH segmentations were generated using native segmentation tools and errors were corrected by deleting, fusing, or fragmenting incorrectly segmented cells. Positive cells were identified by thresholding and manually reviewed based on FISH signal. Imaris (Oxford Instruments) was used for the processing and quantification of multi-z plane confocal images. Cell surfaces were determined using DAPI signal and identified by FISH signal thresholding. Detection errors were manually corrected by reclassifying segmented surfaces based on FISH signal.

## Identification and classification of specialized neoblasts by MERFISH

To identify individual neoblasts, we utilized 13 known neoblast-enriched markers (Supplementary Table 1) and assessed their expression around individual nuclei. Calls were determined by the manual counting and assignment of neoblast transcripts to individual nuclei contained within the parenchymal region of the animal. To determine the threshold for positive calls, ~100 *smedwi-1*+ cells (defined as cells with at least 1 detectable *smedwi-1* transcript) were randomly assessed and quantified for neoblast marker transcript levels (including *smedwi-1*). Nuclei identified as neoblasts demonstrated robust enrichment and co-expression of known neoblast genes, and a minimal transcript threshold was set at ≥5 *smedwi-1* transcripts and ≥6 additional neoblast marker transcripts (determined by median splitting). On average, positively called neoblasts expressed 8 *smedwi-1* transcripts and 37 additional neoblast transcripts whereas negative calls expressed on average 1 *smedwi-1* transcript and 8 additional neoblast transcripts (Supplementary Fig. 6b, c). For images in Fig. 5c and Supplementary Fig. 28, *smedwi-1* single molecule FISH was utilized to identify neoblasts. For comparisons of *smedwi-1* low and high cells, *smedwi-1* transcripts were quantified and calls classified as described above.

To identify individual specialized neoblasts, neoblasts were manually assessed for the enriched expression of FSTFs. Calls were made by using the following criteria: 1. the co-expression of multiple known FSTFs in single cells as a fate signature; 2. the enrichment of a single class of FSTFs within the cell (>50% FSTF single class composition). Any cells with a low FSTF transcript count (<5) or no clear enrichment for a single class of FSTFs were not classified.

## Distance measurements to mature tissues by MERFISH

To determine the distance of specialized neoblasts to different mature tissues in MERFISH experiments, the centroid point of each specialized neoblast classified nucleus was measured to the nearest mature anatomy defined as a nucleus centroid positive for tested mature tissue markers and positioned at stereotypical anatomical locations. In brief, epidermal cell comparisons were made to *PRSS12*+ cells at the outer edge of the animal; intestinal cell comparison were made to *mat*+ cells at defined gut branches; neural cell comparison were made to the ventral nerve cords, cephalic ganglia, or peripheral neural cells; protonephridial cell comparisons were made to flame cells, proximal tubules, or distal tubules; muscle cell comparisons were made to intestinal muscle, dorsoventral muscle, or body-wall muscle; pigment cell comparisons were made to body pigment cells on the periphery of the animal; parenchymal cell comparisons were made to dd4762, *fer3l-2, ZAN6, glipr-1, X1.A.B7.1, mag-1, FAM115C-like,* dd829, dd385, and *SSPO* positive cells within the parenchymal space.

For measurements in regenerating contexts, neoblasts at injuries were classified as described above. Distance measurements were made to the closest mature tissues in the wound site (old tissue) or newly forming blastema (new tissue) using the methodology described above. In sagittal wounds, distance measurements were made to the

closest mature tissues of identified ventral neural specialized neoblasts in pre-existing tissue ("old") or wound tissue ("new") to mature neurons of the ventral nerve cord (identified by characteristic anatomical positioning and clustering of mature neurons on the ventral side of the animal) and to other mature cell types as described above.

### Distance measurements to mature tissues by FISH

For the measurement of pharyngeal specialized neoblast to the pharynx, *foxA+/smedwi-1+* pharyngeal specialized neoblasts were identified by FISH in uninjured planarians. Because all pharyngeal progenitors incorporate into the pharynx through a singular connection (esophagus) located at the anterior of the retracted pharynx (and do not traverse the pharyngeal cavity), distances were calculated using the following methodology: 1) the body was divided into 8 regions divided by borders set at the edges of the pharyngeal cavity and pharynx-body connection; 2) straight-line distance measurements were determined by measuring from the identified specialized neoblast to the nearest anterior border intersection point or to the pharynx-body connection if closer; 3) If measurements were made to border intersection points, additional distance measurements were made following border edges until contact is made with the pharynx-body connection, representing the shortest possible distance to the esophagus.

For measurement of *ovo+/smedwi-1+* eye-specialized neoblasts to the eye, eyes were modeled in 3-dimensions using Imaris, and straight-line distances from identified eye-specialized neoblasts to the closest surface of the eye were measured.

For the measurement of eye-specialized neoblasts and pharyngeal specialized neoblasts to different mature tissues, straight-line distances in 3-dimensions were made to the closest surface of the epidermis, intestine, and central nervous system (cephalic ganglia and ventral nerve cords) identified by DAPI staining and anatomical position.

For EdU+/*smedwi-1-* distance measurements, uninjured animals were pulsed with EdU for 18 h and fixed after 7 days. EdU+/*smedwi-1-* cells were identified by FISH in a region anterior to the base of the brain, and straight-line distances to the nearest *smedwi-1+* neoblast were measured.

### Voronoi analysis

Voronoi neighborhood analysis was performed on 3-dimensional neoblast positions extracted from Imaris sample processing. The freud Python package was used to generate Voronoi polygons for points in the sample space and identify boundary relations between polygons. Voronoi polygons that share a boundary were identified as neighbors, and neighborhoods (list of all neighbors) were characterized for every cell in the space. Neoblast class composition of the neighborhoods for all cells in a sample was determined by identifying the percentage of neighbors of respective neoblast classes. Neighborhood class compositions for each neoblast class identified in a sample were then plotted as binned histograms for all cells of a given neoblast query class.

### Nearest neighbor analysis and randomized nearest neighbor simulations

Nearest neighbor analysis was performed on 3-dimensional neoblast positions extracted from Imaris sample processing. Nearest neighbors were identified by calculating the Euclidian distance between all cells in a sample and determining, for every cell, which other cell in the space was closest. For all cells of a given specialized neoblast class, the class of their identified nearest neighbors was determined. The percentage of nearest neighbors for all cells of a specialized neoblast class that corresponded to the respective classes present in the sample were identified, and nearest neighbor class composition was plotted as a stacked barchart for all cells of a given neoblast query class. Type labels associated with neoblast positions were

randomly shuffled using the MATLAB randperm function 1000 times, and nearest neighbor analysis performed for each randomized position run. Nearest neighbor class composition for each randomized position run was plotted as binned histograms for all cells of a given neoblast query class.

### Quantification and statistical analysis

Statistical analyses were performed using GraphPad Prism software. Statistical tests and other relevant information are provided in corresponding figure legends.

### Reporting summary

Further information on research design is available in the Nature Portfolio Reporting Summary linked to this article.

## Data availability

All data supporting the findings of this study are available in the article or Supplementary Information. Source data are provided with this paper.

## Code availability

Python libraries applied in Voronoi analysis are publicly available and their usage described in the Methods section. Nearest neighbor analysis and simulations were performed in MATLAB as described in the Methods section. Scripts necessary to reproduce these analyses are hosted at: [https://doi.org/10.5281/zenodo.10039363].

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

## Acknowledgements

We thank all members of the Reddien lab for discussions and comments on this work. We thank C. McQuestion for her help in shielded irradiation experiments and C. Rodriguez for his help in running simulations on a local server. P.W.R. is an investigator of HHMI and an associate member of the Broad Institute. The authors acknowledge financial support from NIH R35GM145345 (P.W.R.), NIH T32GM007753 (K.E.O.), and NIH T32GM144273 (K.E.O.).

## Author contributions

C.P., G.M.V., K.E.O., P.W.R. designed the study; C.P, G.M.V., K.E.O. carried out experiments; C.P. designed and carried out MERFISH experiments; G.M.V. performed Voronoi analysis and nearest neighbor simulations; C.P., G.M.V., K.E.O. analyzed data; C.P, G.M.V., K.E.O., P.W.R. wrote and edited the manuscript.

## Competing interests

The authors declare no competing interests.
