## [Peer Review File · Nature Communications]

REVIEWER COMMENTS

Reviewer #1 (Remarks to the Author):

In this article, the authors explore fate specification in the highly regenerative planarian model. In planarians, pluripotent stem cells give rise to dozens and dozens of cell types in both injury and non-injury contexts. Open questions in the field include how adult pluripotent stem cells are regulated and how they choose from such a wide variety of cellular fates. The authors use a new imaging approach, MERFISH, to explore the spatial distribution of fate choice of planarian stem cells and find that stem cells become a diverse set of progenitors in an intermixed spatial organization. Through their careful descriptive analysis, the authors provide intriguing evidence that can be used to argue against a model in which extrinsic (local or regional) cues cause local biases toward specific stem cell fates.

Strengths of the manuscript include application of a novel method and comprehensive analysis to examine several stem cell fates simultaneously. One potential criticism is that the manuscript is descriptive rather than mechanistic. However, the descriptive data generated in this paper certainly provide important information in understanding how planarian stem cells work (and more broadly how adult pluripotent stem cells work in general). Further, this work represents a consequential step forward that is likely to open new doors and lead to new hypotheses. I conclude that it is worthy of publication in Nature Communications, with some changes. Several suggestions follow that the authors can consider in their efforts to solidify their conclusions and to broaden the impact and reach of this paper.

Recommended changes:

1. The manuscript refers to figures Supp. 7-10 in the text and says “FSTF expression signatures corresponded to epidermal, muscle, intestinal, neural, eye, and protonephridial specialized neoblasts (Supplementary Fig. 7-10). Neoblasts without clear fate signatures were unassigned (Supplementary Fig. 8-10).” This summary obscures the observation of some FSTF puncta present in cells that were “called” an alternative fate (e.g., epidermal FSTF in eye progenitors in Supp. Fig. 10). How did the authors determine what level of FSTF expression constituted a “call” for a specific identity/fate and what level constituted an “unassigned” identity? Canonically, each neoblast is thought to only express FSTFs for one identity, but through this method that is not always the case. Is it meaningful or an artifact that there are sometimes transcripts for an FSTF that show up in another fate-specified neoblast? These issues should be brought up in the text, both so that others can employ the technique reproducibly and also because the observations can potentially be informative to the biology of the system. These details also need to be written up very clearly in the methods section.
2. Two tissues that seem to be possible rule-breakers for body-wide intermingling are the pharynx and the eyespot. For both of these tissues, progenitors are regionalized along the body plan and reside mostly nearby the target tissue. However, the authors don’t provide information about the distance of these progenitors from their target (Fig. 1) or quantitative assessment of distance to target tissue (Fig.

3). These tissues should be added to the analysis and any deviations from the trend/main conclusion should be addressed in the text.

3. It seems important that the authors define the way that they assessed target tissues for neurons. Neurons are present in the brain and nerve cords, but also in the peripheral and pharyngeal nervous systems. There may also be as-yet-unstudied neurons in other locations (e.g. an enteric nervous system?). Were neural progenitors' locations defined in relationship to the central nervous system or to the nearest neurons?

4. The authors should clarify how they determined the number of cells needed for the analysis in Fig. 1C and what statistical analyses and comparisons were used to evaluate the data. I agree that the data presented superficially support the claim of the authors but more information is needed to show that the experiments are adequately powered and that the statistical analyses are appropriate.

5. The sorting model is really interesting. In order for the field to move forward and test this model rigorously, it might also be useful for the authors to briefly mention other potential models in the discussion. One alternative model would be that SpNB distant from their target tissues do not become committed and represent the flexible neoblasts described in Raz, et al. I suppose it's also formally possible, though maybe unlikely, that distant neoblasts undergo apoptosis (this might be limited to way out-of-place cells like an eye progenitor in the tail), and only close-enough positioned cells go on to contribute to a tissue.

6. The regeneration data (Fig. 4b) should be quantified and evaluated as rigorously as the homeostasis results. It's hard to tell from single images whether the ratios or relative positions change and quantification/statistical analyses are needed.

Optional changes:

1. There have been reports (including some by the senior author) that smedwi-1^{high} and smedwi-1^{low} cells represent different stages of stem cell differentiation. Can MERFISH quantitatively assess smedwi-1 levels (or stem cell marker levels when they are pooled)? If so, are there differences in spatial distribution for FSTF+ cells based on the level of smedwi-1 expression? This assessment might help to support the author's hypothesis about stem cell migration and provide another layer of mechanism.

2. Do the authors have specific ideas about what the "no ID" neoblasts are? Are these just uncommitted/G0/1 stem cells or stem cells committed to other fates not represented with the given markers? It was interesting that neural progenitors had fewer "no ID" neighbors than

3. TTPA marks only pharyngeal cathepsin+ cells. This should be noted in the text so that the reader understands that other cathepsin+ cell types may not be represented in this analysis.

4. The authors could consider changing the wording "far from their target tissue" since 72 microns is not very far!

Reviewer #2 (Remarks to the Author):

In this manuscript entitled, "Fate specification is spatially intermingled across planarian stem cells", the authors address a longstanding and important question in regeneration biology regarding the generation of differentiated cell types from a multipotent stem cell population. In this case, the stem cells under study are a classical model system for pluripotent stem cells termed neoblasts. Previous work utilizing single-cell RNA-seq suggests that neoblasts can be grouped into subclasses, variously termed "c neoblasts", "alpha", "gamma", "theta" etc. Despite suggestions that these subclasses give rise to different derivatives, almost no follow-up work has addressed the spatial localization of these populations or the mechanisms by which the fates of these subpopulations become restricted. Here's the authors make important observations about the distribution of differentiating neoblasts that have broad implications for their innate potential and for function of adult pluripotent stem cell populations more broadly.

Specifically, the authors utilize multiplexed fluorescent in situ hybridization to spatially localize neoblasts via well-established neoblast markers (Smedwi-1) in combination with fate specific transcription factors (FSTFs) to infer what clusters of neoblasts are fated to become and whether similarly fated neoblasts originate from similar locations in the animal. Importantly, the authors identify that neoblasts fated to become different tissues (e.g. epidermis vs. intestine) may originate side by side, suggesting that clusters of neoblasts related by lineage give rise to different cell types. In addition, a second important finding is that neoblasts that will have similar fates originate from heterogeneous spatial origins. This solves a biological problem that the authors identify in the introduction, that regeneration must allow for the regrowth of body parts from differing regions of the animal.

Additionally, the authors demonstrate that heterogeneous neoblast fate choice is a feature of both homeostatic and regenerating planarians, further bolstering their conclusion that innate features of neoblasts guide cell fate choices in the progeny.

The findings of this paper are of broad interest not only to the planarian field, but more broadly to the field of stem cell biology. Adult pluripotent stem cell populations represent a distinct system from mammalian stem cell populations, which show more highly restricted potential. They are a powerful system to address the maintenance of potential. This study contributes fundamentally to our understanding of how these systems operate. Moreover, this is a clearly constructed and well-written manuscript, whose conclusion are supported squarely by the data presented, and I enthusiastically recommend its publication in Nature Communications.

Specific Comments

“We suggest that coarse and intermingled fate specification followed by progenitor sorting might prove to be a widespread solution to tissue pattern formation challenges, particularly in the context of regeneration.”

This is a compelling conclusion and has broad implications for our understanding of the evolution of stem cell systems.

“We pooled FSTFs into fate expression signatures to improve detection sensitivity and specificity of specialized neoblasts (table S2).”

How can the authors be sure that the pooled signature represent the dynamics of individual cell types represented by more specific markers. E.g. do specific neurons arise from specific spatial domains which could be masked by looking at a mix of neural markers. To distinguish among the possibilities, the researchers should compare the expression of 1 or 2 very robust specific probes to the dynamics of their mixed pool.

“Thus, the specification of some fates was biased towards their target tissue, but occurred broadly and in overlapping fate-specification domains for distinct cell types.”

The bias of some cells toward the tissues to which they will ultimately become is unlikely to represent an origin bias, as the authors demonstrate, but rather, that the cells have already begun migrating to those tissues as they differentiate and perhaps before they have detectable marker expression. However, another possibility is that neoblasts in proximity to differentiated tissues may be biased toward that fate. Evidence supporting or distinguishing between these possibilities is beyond the scope of this article but could be discussed in more detail in the discussion.

The authors claim that neoblasts acquire their fate in a stochastic manner. While choices do indeed appear random, it cannot be ruled out that neoblasts may divide asymmetrically and give rise to cells which will ultimately take on distinct fates in a prescribed order (e.g. the first an epidermal, then intestinal, then muscle etc in a set order). Does the data presented here rule out this possibility?

A few additional points which can be expanded upon here based on the findings and what is known from the literature:

How does this work compare to intermediate cell fate decisions as previously described in the model from Zeng, 2018?

As the authors argue that cell fate decisions are based upon intrinsic properties of the neoblasts, do the authors detect any substructure? Can environmental conditions impact the ratios?

How do the authors confirm that the derivatives of neoblasts expressing FSTFs do indeed give rise to those cell types?

Minor Points

Awkward phrasing

“Our work in conjunction with previous studies identifies central tenets that can facilitate 30 regeneration.”

Please indicate in the legend of figure 1 whether the probes are mixed pools or individual probes as is indicated in the legend of figure 2.

Reviewer #3 (Remarks to the Author):

In the manuscript entitled “Fate specification is spatially intermingled across planarian stem cells”, Park et al. combine multiplexed error-robust fluorescence in situ hybridization (MERFISH) and whole-mount FISH to classify neoblasts according to the expression of specific transcription factors and its probable fate choice. The analysis of their distribution allows the authors to conclude that they are frequently distributed far from their target tissues and in a highly intermingled manner. The authors propose that fate choice involves stem-cell intrinsic processes and that the final position of cells is achieved through migration of intermingled neoblasts types.

The results presented in this manuscript are consistent with previous results of the group reported in Raz et al. 2021, in which they demonstrate that neoblasts can divide asymmetrically and give rise to different cell-type fates. In the present manuscript the authors analyze the expression of multiple FSTF and classify the neoblasts into: epidermal, muscle, intestinal, neural, eye, and protonephridial. The analysis of their distribution in intact animals lead the authors to conclude that specialized neoblasts are intermingled, since the distance to non-self neoblasts is shorter than to self neoblasts. The distance to

the corresponding mature tissue is also not related to the nature of the neoblasts, and the authors conclude that pattern formation is sustained by a process of migration. The study is technically challenging and has allowed the identification of individual specialized neoblasts and a systematic analysis of their distribution, which provides a novel view of planarian stem cells and biology.

However, some major concerns must be addressed before publication. The main ones are:

- The authors refer to the importance of this stem-cell intrinsic properties and the migratory assortment of progenitors to understand planarian regeneration, which is a challenging process because of the arbitrariness of its start point. However, the study is almost focused in the analysis of planarians during homeostasis, while regeneration is superficially studied.

- The conclusions reached seem not to consider objectively all the data obtained or all the data that could be easily obtained, since 1) some tendency to find self neoblasts as neighbors is detected, but it is obviated, and 2) regional specific neoblasts as the ones corresponding to the eyes are not systematically analyzed. On the contrary, muscle, epidermal and intestinal neoblasts are the ones analyzed, but they correspond to tissues that are all around the planarian body.

Specific issues:

- Pag 3 lines 44 and following. The systematic characterization of FSTFs expression allowed the classification of neoblasts into epidermal, muscle, intestinal, neural, eye, and protonephridial. The ones that not were included to any of these categories were not studied. Since all neoblasts express multiple FSTFs (not always corresponding to the same cell type), which were the criteria followed to classify each neoblast in one category? And to classify them as "unassigned"? Please specify how this categorization was performed, since it is the basis of the following results. It must be explained in the methods in detail.

- According to Raz et al. 2021, a neoblast can give rise to different fated cells during mitosis. Then, the specialized neoblast identified in this systematic study, can be really considered specialized if it could be that after the next mitosis its fate changes?

- Fig 1c. How is the distance from neoblasts to differentiated tissues calculated? For instances, neural cells or intestine cells are everywhere. Which are the criteria followed? It must be explained in the methods in detail.

- Specialized neoblasts distribution and EdU incorporation experiments lead the authors to suggest that initial fate choices are made everywhere in the planaria and are not related with any positional information. However, postmitotic cells are already found in specific locations related to differentiated cells. How are mitotic cells rearranged to reach their final position? The authors argue that migration must have an important role. However, migration is only demonstrated in irradiation experiments, which is not the context of homeostasis neither regeneration.

If the existence of cell migration cannot be demonstrated in more physiological, non-irradiated conditions, this proposal should remain in the discussion.

- The authors find that adjacent neoblast showed fate specification for tissues from different embryonic germ layers. It is also commented that this is not the case in most animal embryos. However, 1) the existence of neuromuscular precursors in vertebrate and invertebrate embryos has been reported in

several studies, and 2) the comparison of planarians with embryos is interesting, but it should be rather compared with whole-body regenerating animals, as cnidarians or acoels.

In relation to this issue, several single cell sequence analyses have been published on planarian cells in the last few years. The results presented in this study should be discussed according to these data. For instances, are there found cell types that could correspond to precursors of different germ layers, as suggested in the text?

- The authors find muscular, epidermal and excretory specialized neoblasts in all locations and intermingled. However, these tissues are found all along the planarian body. What about region-specific neoblasts?

Specifically, do the authors find eye neoblasts in the prepharyngeal region of an intact animal? An in the postpharyngeal? And in A or P clones of a sublethal-irradiated animal? And what about regeneration? Do the authors find eye neoblasts in A blastemas and also in P blastemas? Without this specific analysis the conclusions reached in the present study are not supported.

Additional studies that should be performed in this line: if A and P clones in sublethal irradiated animals are analyzed, are there differences in the composition of the clones related with the AP position? This kind of analysis should be performed, since the finding of some differences would mean that positional cues are in fact important to specify cell fate during mitosis.

- In the analysis of Fig3D and Supp 21, the authors identify the neighbors for a given neoblast and conclude that this data supports their previous data and a model in which fate specification occurs in a highly intermingled "salt-and-pepper" distribution. However, looking at the data, it is true that in some cases non-self neoblasts are the more represented neighbors, but in others are the self neoblasts. These cases should be also commented and included in the discussion of the results. Furthermore, the analysis of regional neoblasts as the ones corresponding to the eyes or to the pharynx region should be included in the analysis.

Regarding this data represented in the graphs of Fig3d-e and Supp 21, a better explanation of the graphs or an improvement should be performed to make it easy their interpretation.

As introduced in the manuscript, the biological question behind this study is to understand how stem cells decide their fate during regeneration. However, almost all the analysis are performed in homeostatic animals. The same analysis performed in intact animals should be performed in regenerating blastemas. Are the specialized neoblast closer to their corresponding differentiated structures? Are neural fated neoblasts closed to the regenerating nerve cords? Are intestinal fated neoblasts closed to the regenerating intestine? Are self neoblasts closer between them than to non-self neoblasts? As already pointed out, are eye neoblasts present in P blastemas?

The authors have the tools to answer this key question, which was the initial interest, according to the introductory lines.

- What is the point of the shielded irradiated planarians experiment? The migration of neoblasts in these conditions has already been reported in different studies (Dr. Aziz lab). Furthermore, as already pointed out, if the authors would like to demonstrate the role of migration in positioning differentiated cells, it should be demonstrated in regenerating or intact animals, but not in a context in which the population

of neoblasts is dramatically decreased and migration is the only way of survival. It has been demonstrated in several animal models that cell behaviors is not comparable between physiological and extreme conditions of cell depletion.

Additional issues:

- From supp 7 to 10, Fig 3B, 3F, supp 23... please show the region to which the cells correspond.
- Fig 1A, muscle and pigment cells are difficult to be distinguished.
- Movies are not named as S1, etc...
- Page 6 line 4- "subtotal" should be "sublethal".
- It is hard to understand this manuscript for the general reader. Basic planarian biology should be included: the cellular and molecular basis of planarian regeneration, the stages of regeneration, what is a sublethal irradiation...
- It is difficult to follow the results in the current format. Figures should contain information about the region analyzed, and explained in more detail.
- Supp 24. Circles must be of the same thickness (all thinner).
- Pag4-line 8 "Supplementary Fig. a,b"- number is missing.
- The authors found that specialized neoblasts were closer to parenchymal cells, and in several cases also with intestinal cells. A possible explanation could be the role as niches of these tissues. This issue could be discussed.
- The model proposed in Fig 4d is too simplistic. A more elaborated model, in line with a more elaborated discussion, would enrich the manuscript.

Reviewer #1 (Remarks to the Author):

In this article, the authors explore fate specification in the highly regenerative planarian model. In planarians, pluripotent stem cells give rise to dozens and dozens of cell types in both injury and non-injury contexts. Open questions in the field include how adult pluripotent stem cells are regulated and how they choose from such a wide variety of cellular fates. The authors use a new imaging approach, MERFISH, to explore the spatial distribution of fate choice of planarian stem cells and find that stem cells become a diverse set of progenitors in an intermixed spatial organization. Through their careful descriptive analysis, the authors provide intriguing evidence that can be used to argue against a model in which extrinsic (local or regional) cues cause local biases toward specific stem cell fates.

Strengths of the manuscript include application of a novel method and comprehensive analysis to examine several stem cell fates simultaneously. One potential criticism is that the manuscript is descriptive rather than mechanistic. However, the descriptive data generated in this paper certainly provide important information in understanding how planarian stem cells work (and more broadly how adult pluripotent stem cells work in general). Further, this work represents a consequential step forward that is likely to open new doors and lead to new hypotheses. I conclude that it is worthy of publication in Nature Communications, with some changes. Several suggestions follow that the authors can consider in their efforts to solidify their conclusions and to broaden the impact and reach of this paper.

We thank the reviewer for their support and assessment of this work.

Recommended changes:

1. The manuscript refers to figures Supp. 7-10 in the text and says “FSTF expression signatures corresponded to epidermal, muscle, intestinal, neural, eye, and protonephridial specialized neoblasts (Supplementary Fig. 7-10). Neoblasts without clear fate signatures were unassigned (Supplementary Fig. 8-10).” This summary obscures the observation of some FSTF puncta present in cells that were “called” an alternative fate (e.g., epidermal FSTF in eye progenitors in Supp. Fig. 10). How did the authors determine what level of FSTF expression constituted a “call” for a specific identity/fate and what level constituted an “unassigned” identity? Canonically, each neoblast is thought to only express FSTFs for one identity, but through this method that is not always the case. Is it meaningful or an artifact that there are sometimes transcripts for an FSTF that show up in another fate-specified neoblast? These issues should be brought up in the text, both so that others can employ the technique reproducibly and also because the observations can potentially be informative to the biology of the system. These details also need to be written up very clearly in the methods section.

We now include more detail about the methodology applied to identify specialized neoblasts by MERFISH (new Methods section titled “Identification and classification of specialized neoblasts by MERFISH”).

Whereas some FSTFs are highly specific to a specialized class (e.g., *ovo* for eye specialized neoblasts), others are found to be enriched but not exclusive to a single class (e.g. *zfp-1*, *prox-1*, *eya*). This could be related to roles in multiple fates, or biological or technical noise. These observations are consistent with observations using other methods. For example, the expression of *zfp-1*, while enriched in epidermal specialized neoblasts, is also expressed to a lower degree in other classes including intestinal specialized neoblasts (Fincher et al. 2018). We have added in more description to clarify and contextualize these observations.

2. Two tissues that seem to be possible rule-breakers for body-wide intermingling are the pharynx and the eyespot. For both of these tissues, progenitors are regionalized along the body plan and reside mostly nearby the target tissue. However, the authors don’t provide information about the distance of these progenitors from their target (Fig. 1) or quantitative assessment of distance to target tissue (Fig. 3). These tissues should be added to the analysis and any deviations from the trend/main conclusion should be addressed in the text.

Specialized neoblasts of both of these classes have now been analyzed by whole-mount FISH. We assessed the distance of *ovo*+ eye specialized neoblasts and *foxA*+ pharyngeal specialized neoblasts to their respective target tissues, and to different mature tissues including the intestine, epidermis, and central nervous system (brain and ventral nerve cords) which are detectable by DAPI staining and anatomical position (Figure Fig 3h,i). We describe the methodology in Materials and Methods (See “Distance measurements of specialized neoblasts to mature tissues by FISH”).

We find that like other tested fates, eye- and pharynx-specialized neoblasts are located closer to different mature tissues than to their target tissue (Fig 3h,i). We also analyzed the neoblast neighborhoods of these specialized neoblast classes, and found them to be highly heterogeneous, like other analyzed classes. In many cases, for example, eye specialized neoblasts had no immediate eye-specialized neoblast neighbors. These data have been added to Fig. 3j,k, and Supplementary Fig. 24.

Importantly, we don't view these two progenitor types as rule breakers so much as indicative of the fact that global patterning has influence on choices available to neoblasts, yet at the local level the pattern is highly intermingled. To further emphasize this aspect of the model we added more discussion and a new panel to the model figure depicting the concept that regional differences in choices exist, and intermingling is pervasive (Fig. 7b).

3. It seems important that the authors define the way that they assessed target tissues for neurons. Neurons are present in the brain and nerve cords, but also in the peripheral and pharyngeal nervous systems. There may also be as-yet-unstudied neurons in other locations (e.g. an enteric nervous system?). Were neural progenitors' locations defined in relationship to the central nervous system or to the nearest neurons?

Neural progenitors were measured relative to defined neural structures including the cephalic ganglia and ventral nerve cords. Distances to pharyngeal neural populations were not measured (because the analyzed structures represent the closest anatomically defined neural populations, and pharynx neurons are much further away). We now note specific methodology utilized in the Methods section titled "Distance measurements of specialized neoblasts to mature tissues by MERFISH". We included a pan-neural marker in the MERFISH data, and no robust, clear enteric nervous system was visible, nor has such a population been revealed in planarian cell-type atlas investigations (Supplementary Fig. 4b, Fincher 2018). It is conceivable that unstudied neural populations are positioned closer to particular neural specialized neoblasts are found, but we performed additional analyses to bolster the conclusion that neural specialized neoblasts are specified broadly and frequently distant from at least major target tissues:

First, we studied a more specific neural population with defined mature tissue markers and known FSTFs – serotonergic neurons. Mature serotonergic neurons express *sert* whereas specialized neoblasts can be detected by expression of *pitx* (Marz et al. 2013, Currie and Pearson 2013). We first identified mature serotonergic neurons through the expression of *sert* (Supplementary Fig. 12a). We identified 17 neoblasts positive for *pitx* within the anterior half of the tissue section and measured their distances to mature tissues, including non-serotonergic neural cell types (Supplementary Fig. 12b-d). Serotonergic neuron specialized neoblasts were not located significantly closer to mature serotonergic neurons compared to other neural classes, indicating that specification of these neurons is not spatially biased to their target tissue (Supplementary Fig. 12f). Serotonergic neural specialized neoblasts were also located closer to parenchymal cell types than to mature serotonergic neurons, consistent with prior results (Supplementary Fig. 12e). Second, we assayed proximity of neural specialized neoblasts to another population of neurons not analyzed in the original submission, dorsal peripheral neurons (Supplementary Fig. 12g). Consistent with prior findings, dorsal neural specialized neoblasts were located closer to parenchymal cell types than to mature dorsal neurons and were a similar distance or closer to multiple other differentiated cell types (Supplementary Fig. 12g,h).

4. The authors should clarify how they determined the number of cells needed for the analysis in Fig. 1C and what statistical analyses and comparisons were used to evaluate the data. I agree that the data presented superficially support the claim of the authors but more information is needed to show that the experiments are adequately powered and that the statistical analyses are appropriate.

In comparisons of the distance of queried specialized neoblasts to either parenchymal cell types or target mature tissues, we applied two-sided Welch's t tests for each of the tested fates (Supplementary Fig. 11d). We find that specialized neoblasts of all tested fates are significantly closer to parenchymal cell types than to their respective target tissues. The overall points of this data are general: specialized neoblasts of different fates are broadly distributed, they are frequently far from their target tissues, and their distances to target tissues are similar and, for individual cells, often closer to other tissues.

5. The sorting model is really interesting. In order for the field to move forward and test this model rigorously, it might also be useful for the authors to briefly mention other potential models in the discussion. One alternative

model would be that SpNB distant from their target tissues do not become committed and represent the flexible neoblasts described in Raz, et al. I suppose it's also formally possible, though maybe unlikely, that distant neoblasts undergo apoptosis (this might be limited to way out-of-place cells like an eye progenitor in the tail), and only close-enough positioned cells go on to contribute to a tissue.

The specialized neoblasts described in Raz et al. 2021 represent S/G2/M phase neoblasts committing to their respective fates. About half of the divisions of neoblasts have an asymmetric outcome with one daughter leaving the neoblasts state and becoming a post-mitotic progenitor that will differentiate (Raz et al. 2021). Some neoblast divisions can also produce two differentiating daughter cells. More than half of identified specialized neoblasts in this study would thus be expected to produce at least one post-mitotic daughter cell that differentiates according to that fate. New fates can be selected in the division of a daughter neoblast. At this point, we have not observed evidence of fate switching in the course of a single division (hybrid states clearly indicative of switching in S/G2 have not been observed); instead the specialization state, if anything, becomes more mature as a cell progresses in the cell cycle (van Wolfswinkel et al. 2014).

It is unlikely that distant specialized neoblasts selectively undergo apoptosis. If this were to be the case, it would be expected that neoblasts would contribute a significant portion to the homeostatic rate of cell death observed within planarians given the abundance of these cells (neoblasts are ~15% of all cells and a majority are specialized in spatially broad regions in this study and prior work, van Wolfswinkel et al. 2014, Fincher et al. 2018). However, no difference in total apoptosis was observed after neoblast depletion, suggesting that neoblasts do not contribute significantly to the overall population of homeostatic TUNEL+ cells (Pellettieri et al. 2010). In colony analyses (Wagner et al. 2012), all growing colonies make post-mitotic epidermal progenitors regardless of location. Furthermore, under alternative models involving distant apoptosis or fate switching of distant cells, we would expect the distribution of fate choice to exhibit more neighbor composition structure and more spatial bias for specialized neoblasts towards target tissues (e.g., if distal specialized neoblasts were pruned or switched into other states).

The shielded irradiation experiments demonstrate that specification can occur far from target tissues, and that post-mitotic progeny can be targeted to correct anatomical locations (Fig. 6c). It has previously been demonstrated that neoblasts are not highly migratory in homeostatic conditions but that post-mitotic epidermal progenitors are able to move far from the neoblast containing region (Guedelhofer and Sánchez Alvarado 2012, Abnave et al. 2017). Migration has also been observed in non-irradiated contexts, for example, eye-specialized neoblasts produce post-mitotic progenitors far from the eye and distant cells in eye regeneration initiate expression of markers for differentiated eye cell types (e.g., tyrosinase of the optic cup) indicating these cells are likely to incorporate into the eye itself (Lapan and Reddien 2011; Lapan and Reddien 2012). Experiments utilizing lethal doses of irradiation, which prevents new cell production, resulted in transient growth of the eyes by approximately the number of cells present within eye progenitor trails during regeneration suggesting that even distal cells contribute to the eye (Lapan and Reddien 2011). Migration in non-irradiated conditions has also been observed for the epidermis (Eisenhoffer et al. 2008, Tu et al. 2015) and must occur for the pharynx because the pharynx lacks neoblasts and undergoes new cell incorporation and regeneration (Adler et al. 2014; LoCascio et al. 2017, Bohr et al. 2021). BrdU labeling shows rapid incorporation into the head tip, from which neoblasts are excluded (Newmark, 2000). Finally, inhibition of migration (through the application of β -integrin RNAi) results in the formation of aberrant neural ectospheres associated with the inability for progenitors to be targeted to their target tissues, likely near the sites in which these cell types are specified within the anterior of the animal (Bonar and Petersen 2017, Seebeck et al. 2017). We also expanded our analysis by assessing migration in non-irradiated conditions. We pulsed uninjured animals with EdU and assessed the incorporation of new progeny into mature tissues of the anterior head tip, a region devoid of neoblasts; thus, any new incorporation into these tissues must have occurred initially within distant neoblasts, then migrated to their target. After pulsing with EdU, we observed robust movement of post-mitotic EdU+ cells into the head tip, beyond the region containing neoblasts (Supplementary Fig. 33a,b). We observed incorporation of EdU into anterior-most *cintillo*+ sensory neurons and anterior-most *collagen*+ muscle cells (Fig. 6d,e). Similarly, we pulsed animals with EdU 4 days post decapitation and assessed incorporation into the blastema, again a region excluding neoblasts, at 6dpa. We observed incorporation of EdU into the anterior regions of the brain, indicating that a neural cell specified at the wound site had migrated into the anterior regions of the blastema (Fig. 6f, Supplementary Fig. 33d). It remains possible that some bias in productive differentiation occurs in cells

closer to their target tissue. However, taken together, these data all support the model that fate specification can occur at distant sites, and that post-mitotic progenitors migrate to their target destinations.

We expanded our discussion of potential alternative models and the arguments supporting migratory sorting in the revised discussion section of the paper.

6. The regeneration data (Fig. 4b) should be quantified and evaluated as rigorously as the homeostasis results. It's hard to tell from single images whether the ratios or relative positions change and quantification/statistical analyses are needed.

We performed the suggested additional analyses of regenerating wound sites. Using MERFISH, we mapped the positions of specialized neoblasts in anterior and posterior facing wounds at 72hpa and 96hpa, respectively, and also analyzed regeneration at sagittal wounds (Fig. 5b,f, Supplementary Fig. 28). We found that specialized neoblasts are highly intermingled in anterior, posterior, and sagittal regeneration, including regionally specified eye specialized neoblasts at anterior-facing wounds. To further characterize fate choice in regeneration, we assessed neoblast neighborhoods of specialized neoblasts at anterior-facing wounds at 72hpa in pairwise FISH comparisons (Fig. 5d and Supplementary Fig. 29). Neighborhoods were heterogenous and often contained specialized neoblasts identified for a different fate, consistent with measurements from uninjured animals. Neoblast neighborhoods around eye-specialized neoblasts were also highly heterogeneous, with many such neoblasts possessing no immediate other eye-specialized neoblast neighbors (Fig. 5e and Supplementary Fig. 29a, d). Furthermore, we measured the distances of identified specialized neoblasts at wound sites to different mature tissues, including their respective target tissue in the case of sagittal regeneration, and find that specialized neoblasts are located closer to parenchymal cell types than to other mature or target tissues at wounds and blastemas (Fig. 5b,c,f,g and Supplementary Fig. 30). Taken together, these results indicate that fate choice in regeneration occurs in a spatially heterogeneous manner and does not occur directly adjacent to newly forming target tissues.

Optional changes:

1. There have been reports (including some by the senior author) that *smedwi-1*high and *smedwi-1*low cells represent different stages of stem cell differentiation. Can MERFISH quantitatively assess *smedwi-1* levels (or stem cell marker levels when they are pooled)? If so, are there differences in spatial distribution for FSTF+ cells based on the level of *smedwi-1* expression? This assessment might help to support the author's hypothesis about stem cell migration and provide another layer of mechanism.

We tried the suggestion, thank you. We classified *smedwi-1*+ cells as either *smedwi-1* high or *smedwi-1* low (see Methods) and measured their distances to different mature tissues. We found that *smedwi-1* low cells were indeed located closer to target tissues than *smedwi-1* high cells and these new data were added to the paper (Supplementary Figure 32).

2. Do the authors have specific ideas about what the "no ID" neoblasts are? Are these just uncommitted/G0/1 stem cells or stem cells committed to other fates not represented with the given markers? It was interesting that neural progenitors had fewer "no ID" neighbors than

In our analyses, no ID neoblasts were those that were negative for all tested FSTFs. Uncommitted, G1, or positive for untested FSTF are all possibilities. We added a note to this effect in the text. The neuronal specialized neoblasts are an interesting population for future studies. Neural specialized neoblast neighborhoods still exhibited a high degree of heterogeneity in observed neighbor composition, consistent with other tested fates. We anticipate that future work further dissecting neuronal specification to identify specific gene expression programs may provide the tools to assess the spatial distribution of subsets of these cell types in fine detail (beyond the scope of this work).

3. TTPA marks only pharyngeal cathepsin+ cells. This should be noted in the text so that the reader understands that other cathepsin+ cell types may not be represented in this analysis.

We modified the text to note the pharyngeal specificity of this marker.

4. The authors could consider changing the wording “far from their target tissue” since 72 microns is not very far!

We modified the wording describing the distance measurements.

Reviewer #2 (Remarks to the Author):

In this manuscript entitled, “Fate specification is spatially intermingled across planarian stem cells”, the authors address a longstanding and important question in regeneration biology regarding the generation of differentiated cell types from a multipotent stem cell population. In this case, the stem cells under study are a classical model system for pluripotent stem cells termed neoblasts. Previous work utilizing single-cell RNA-seq suggests that neoblasts can be grouped into subclasses, variously termed “c neoblasts”, “alpha”, “gamma”, “theta” etc. Despite suggestions that these subclasses give rise to different derivatives, almost no follow-up work has addressed the spatial localization of these populations or the mechanisms by which the fates of these subpopulations become restricted. Here’s the authors make important observations about the distribution of differentiating neoblasts that have broad implications for their innate potential and for function of adult pluripotent stem cell populations more broadly.

Specifically, the authors utilize multiplexed fluorescent in situ hybridization to spatially localize neoblasts via well-established neoblast markers (Smedwi-1) in combination with fate specific transcription factors (FSTFs) to infer what clusters of neoblasts are fated to become and whether similarly fated neoblasts originate from similar locations in the animal. Importantly, the authors identify that neoblasts fated to become different tissues (e.g. epidermis vs. intestine) may originate side by side, suggesting that clusters of neoblasts related by lineage give rise to different cell types. In addition, a second important finding is that neoblasts that will have similar fates originate from heterogeneous spatial origins. This solves a biological problem that the authors identify in the introduction, that regeneration must allow for the regrowth of body parts from differing regions of the animal.

Additionally, the authors demonstrate that heterogeneous neoblast fate choice is a feature of both homeostatic and regenerating planarians, further bolstering their conclusion that innate features of neoblasts guide cell fate choices in the progeny.

The findings of this paper are of broad interest not only to the planarian field, but more broadly to the field of stem cell biology. Adult pluripotent stem cell populations represent a distinct system from mammalian stem cell populations, which show more highly restricted potential. They are a powerful system to address the maintenance of potential. This study contributes fundamentally to our understanding of how these systems operate. Moreover, this is a clearly constructed and well-written manuscript, whose conclusion are supported squarely by the data presented, and I enthusiastically recommend its publication in Nature Communications.

We thank the reviewer for their support and assessment of this work.

Specific Comments

“We suggest that coarse and intermingled fate specification followed by progenitor sorting might prove to be a widespread solution to tissue pattern formation challenges, particularly in the context of regeneration.”

This is a compelling conclusion and has broad implications for our understanding of the evolution of stem cell systems.

We were happy to see the positive impression.

“We pooled FSTFs into fate expression signatures to improve detection sensitivity and specificity of specialized neoblasts (table S2).”

How can the authors be sure that the pooled signature represent the dynamics of individual cell types represented by more specific markers. E.g. do specific neurons arise from specific spatial domains which could be masked by looking at a mix of neural markers. To distinguish among the possibilities, the researchers should compare the expression of 1 or 2 very robust specific probes to the dynamics of their mixed pool.

We added several new single marker analyses. First we added analysis of *ovo+* eye progenitors with *soxP-3+* epidermal progenitors. We added images of neighboring eye -epidermal specialized neoblast neighbors. We also performed neighborhood analyses for this experiment and added that data, which showed highly

heterogeneous neighborhoods (Figure 3j, 5e and supplement 24 and 29). We performed similar analyses using *foxA* as a single probe for the pharynx and *soxP-3* for the epidermis (Figure 3k and supplement 24). We selected FSTFs for the work involving probe pools that demonstrated high lineage specificity. In the case of the epidermis, specialized neoblasts were detected throughout using a single marker, *soxP-3*, which shows high specificity and signal strength for epidermal specialized neoblasts. In some cases, probe pools were needed for sufficient signal strength for cell calls. For MERFISH experiments, combinatorial expression was utilized as a means to increase the confidence of specialized neoblast identity calls, and all cases demonstrated the expression of known canonical FSTFs. Some FSTFs used in combinatorial calls (signatures) were not exclusive to a particular specialized neoblast class but were still enriched in a single class (Fincher et al. 2018). We also added multiple additional examples of neighboring neoblasts pairs of very different fates using single markers in the MERFISH data, including: *hnf4* (intestine) – *gata4/5/6-2* (DV muscle); *hnf4* – *myoD* (longitudinal muscle); *gata4/5/6-1* (intestine) – *pou2/3* (protonephridia); *gata4/5/6-2* – *pou2/3*; *p53* (epidermis) – *hnf4* (intestine) (Supplemental 20b).

In the case of neuronal cell types, a number of reported neuronal FSTFs identify broad populations that can produce many different mature neuronal cell types, where the diversity of the mature neural populations is greater than at the specialized neoblast level (Scimone et al. 2014, Fincher et al. 2018). We anticipate that future work further dissecting neuronal specification to identify specific gene expression programs might provide the tools to assess the spatial distribution of these cell types in finer detail (beyond the scope of this work). To better understand specific neural dynamics, we studied specification of serotonergic neurons. Mature serotonergic neurons express *sert* whereas specialized neoblasts can be detected by expression of *pitx* (Marz et al. 2013, Currie and Pearson 2013). We first identified mature serotonergic neurons through the expression of *sert* (Supplementary Fig. 12a). We then identified 17 neoblasts positive for the single marker *pitx* within the anterior half of the tissue section and measured their distances to mature tissues, including non-serotonergic neural cell types (Supplementary Fig. 12b-d). We find that serotonergic neural specialized neoblasts are located closer to parenchymal cell types than to mature serotonergic neurons, consistent with prior results (Supplementary Fig. 12e). Importantly, serotonergic neuron specialized neoblasts were not located significantly closer to mature serotonergic neurons compared to other neural classes, or other non-neural tissues, indicating that specification of these neurons is not spatially biased to their target tissue (Supplementary Fig. 12f). Additionally, we assayed another population of neurons, dorsal peripheral neurons, and found that dorsal neural specialized neoblasts are significantly closer to parenchymal cell types than to dorsal peripheral neurons, consistent with previous results (Supplementary Fig. 12g,h).

“Thus, the specification of some fates was biased towards their target tissue, but occurred broadly and in overlapping fate-specification domains for distinct cell types.”

The bias of some cells toward the tissues to which they will ultimately become is unlikely to represent an origin bias, as the authors demonstrate, but rather, that the cells have already begun migrating to those tissues as they differentiate and perhaps before they have detectable marker expression. However, another possibility is that neoblasts in proximity to differentiated tissues may be biased toward that fate. Evidence supporting or distinguishing between these possibilities is beyond the scope of this article but could be discussed in more detail in the discussion.

We agree with the reviewer that this is an interesting subject to investigate. Our data cannot with certainty distinguish between the spatial bias of some specialized neoblasts resulting from cell movement or a bias in the location of fate choice. However, prior studies have determined that neoblasts are not highly mobile in homeostatic conditions making it unlikely that neoblasts are undergoing large spatial movements towards target tissues, although a small degree of movement may be present (Guedelhofer and Alvarado 2012, Abnave et al. 2017, shielded irradiation experiments in this paper). We have expanded the text to account for this possibility.

We have also expanded our analysis by assessing migration in non-irradiated conditions. We pulsed uninjured animals with EdU and assessed the incorporation of new progeny into mature tissues of the anterior head tip, a region devoid of neoblasts; thus, any new incorporation into these tissues must have occurred initially within distant neoblasts, then migrated to their target. After pulsing with EdU, we observed robust movement of post-mitotic EdU+ cells into the head tip, beyond the region containing neoblasts (new

Supplementary Fig. 33). We observed incorporation of EdU into anterior-most *cintillo*⁺ sensory neurons and anterior-most *collagen*⁺ muscle cells (new Fig. 6d,e). Similarly, we pulsed animals with EdU 4 days post decapitation and assessed incorporation dynamics into the blastema, again a region excluding neoblasts, at 6dpa. We observed incorporation of EdU into the anterior regions of the brain, indicating that a neural cell specified at the wound site had migrated into the anterior regions of the blastema (new Fig. 6f). These data suggest that progeny migration, rather than migration of neoblasts, represents the major driver of tissue patterning.

The authors claim that neoblasts acquire their fate in a stochastic manner. While choices do indeed appear random, it cannot be ruled out that neoblasts may divide asymmetrically and give rise to cells which will ultimately take on distinct fates in a prescribed order (e.g. the first an epidermal, then intestinal, then muscle etc in a set order). Does the data presented here rule out this possibility?

We expanded the discussion section to account for this potential mechanism. We cannot formally exclude the possibility that neoblast fate specification may occur in a set order as described by the reviewer. However, prior work makes a highly prescribed order, as described, less likely. Clonal colony analysis, in which a single surviving neoblast produces a small neoblast colony related by lineage, produces colonies containing varying numbers of epidermal specialized neoblasts (Raz et al. 2021). This phenomenon was also observed for other fates including *foxA*⁺ pharynx specialized neoblasts, *hnf4*⁺ intestinal specialized neoblasts, *foxF-1*⁺ muscle specialized neoblasts, and *prox-1*⁺ intestine/neural specialized neoblasts (Raz et al. 2021). Four cell colonies were analyzed and the frequencies of colonies having 0, 1, 2, 3, or 4 of the particular neoblast classes were determined; the distribution of these classes matched what would be expected from random sampling, given the overall frequency of the neoblast class in the total neoblast population. If there were a stereotyped order of fate choice, we would expect deviation from random (e.g., one might expect a higher frequency of the 0 class to account for all of the instances when the 4 cell colonies were in a different part of some stereotyped order). Because neoblast class frequencies are very different, the prescribed order model would require some states to appear at multiple positions in the order, or to be repeated for different numbers of times, depending on the state and the desired overall output rate. These considerations might also predict the class distribution in 4-cell colonies would not match random predictions. Furthermore, in two cell clones where the mother had an epidermal fate, the daughter neoblast did not always have an epidermal fate or a non-epidermal fate (inconsistent order for this single outcome) (Raz et al. 2021). Finally, neighborhood analyses of neoblast fates (this work) did not reveal obvious signs of order (e.g., always having a particular type of nearest neighbor). Instead, neoblasts could have nearest neighbors of any queried type and the neighborhoods did not reveal overt patterns. That said, there could be a more complex temporal order involving repeated states (to get different overall frequencies) and perhaps some frequency of skipping states such that the numbers appear to match random sampling in available colony data. We therefore note in the discussion that some order to fate transitions is possible and that this will be an important target for future analyses.

A few additional points which can be expanded upon here based on the findings and what is known from the literature:

How does this work compare to intermediate cell fate decisions as previously described in the model from Zeng, 2018?

Our work does not discount the presence of a fully uncommitted clonogenic neoblast (cNeoblast), such as proposed in Zeng et al. 2018. However, prior work from our lab demonstrated that a specialized neoblast that produced a committed post-mitotic daughter of one fate is able to produce a daughter neoblast that is able to produce progeny of a different fate, and that this can occur within a single cell division (Raz et al. 2021). Furthermore, we found that *tgs-1*⁺ neoblasts (described as clonogenic in Zeng et al. 2018) are enriched in known neural FSTFs, and analyses of early neoblast colonies after sublethal irradiation show that *tgs-1*⁺ neoblasts are absent from more than half of assessed colonies (at the frequency expected for random sampling of neural fates in colonies) (Raz et al. 2021). Similarly, experiments in which a host planarian is lethally irradiated, clearing all neoblasts, and then transplanted with a single neoblast that can then colonize the host, resulted in the formation of colonies at twice the frequency expected than if only *tgs-1*⁺ neoblasts were capable of forming a colony (Raz et al. 2021). These data indicate that specialized neoblasts are capable of self-renewal and that neoblast progeny are able to produce cells of different fates. Under this model, an

intermediate neoblast lineage transition state as described by Zeng et al. 2018 would neither be expected nor required for maintenance of neoblasts and the production of multiple differentiated tissues.

We added additional discussion regarding the potential presence or absence of clonogenic neoblasts in planarians.

As the authors argue that cell fate decisions are based upon intrinsic properties of the neoblasts, do the authors detect any substructure? Can environmental conditions impact the ratios?

The study of cell fate decisions under different environments (feeding, starvation) and conditions (homeostasis, injury) are areas of great interest, although some of these topics were beyond the scope of this work. For example, multiple studies are interested in the concept of regenerative feedback, in which damage to or the absence of a specific tissue can trigger a targeted regenerative response or not, and this is an active area of study in the lab (Tu et al. 2015, LoCasio et al. 2017, Bohr et al. 2021). We added substantial new data in this draft on regeneration. We utilized MERFISH to examine neoblast structure in anterior, posterior, and sagittal regeneration. We also performed quantitative analyses of distances to differentiated tissues at wounds and for some newly forming tissues in blastemas. Finally, we performed quantitative analyses of neoblast neighbors at wound sites in regeneration. We found a high degree of neighborhood heterogeneity and a dispersed nature of neoblast classes in all analyzed regeneration conditions. These new data are presented in Figures 5b-g and Supplementary Figures 28-30.

How do the authors confirm that the derivatives of neoblasts expressing FSTFs do indeed give rise to those cell types?

In many cases, FSTF expression matches the expression signature in a corresponding mature cell type. Depletion of FSTFs by RNAi leads to a corresponding loss of production of their specific mature cell types, which can be detected by histology, secondary markers, and EdU experiments. This depletion occurs during tissue turnover and FSTF RNAi for well studied cases also blocks regeneration of corresponding cell types. Recent single cell RNA-sequencing studies have identified a multitude of FSTFs and corresponding cell types (Fincher et al. 2018) as well as fate trajectories that corroborate prior experimental results (Plass et al. 2018). The role of FSTFs can be exemplified by two well-defined trajectories, for the eye and the epidermis (Lapan and Reddien 2011, 2012, van Wolfswinkel et al. 2014, Tu et al. 2015). In these lineages, transition states from specialized neoblasts to post-mitotic progenitor to mature tissue had well-defined gene markers and FSTFs were required for production of transition states and final tissues.

Minor Points

Awkward phrasing

“Our work in conjunction with previous studies identifies central tenets that can facilitate 30 regeneration.”

We modified the text accordingly.

Please indicate in the legend of figure 1 whether the probes are mixed pools or individual probes as is indicated in the legend of figure 2.

We modified the legend of Figure 1 to clarify whether pools vs individual probes were utilized.

Reviewer #3 (Remarks to the Author):

In the manuscript entitled “Fate specification is spatially intermingled across planarian stem cells”, Park et al. combine multiplexed error-robust fluorescence in situ hybridization (MERFISH) and whole-mount FISH to classify neoblasts according to the expression of specific transcription factors and its probable fate choice. The analysis of their distribution allows the authors to conclude that they are frequently distributed far from their target tissues and in a highly intermingled manner. The authors propose that fate choice involves stem-cell intrinsic processes and that the final position of cells is achieved through migration of intermingled neoblasts types.

The results presented in this manuscript are consistent with previous results of the group reported in Raz et al. 2021, in which they demonstrate that neoblasts can divide asymmetrically and give rise to different cell-type fates. In the present manuscript the authors analyze the expression of multiple FSTF and classify the neoblasts into: epidermal, muscle, intestinal, neural, eye, and protonephridial. The analysis of their distribution in intact animals lead the authors to conclude that specialized neoblasts are intermingled, since the distance to non-self neoblasts is shorter than to self neoblasts. The distance to the corresponding mature tissue is also not related to the nature of the neoblasts, and the authors conclude that pattern formation is sustained by a process of migration. The study is technically challenging and has allowed the identification of individual specialized neoblasts and a systematic analysis of their distribution, which provides a novel view of planarian stem cells and biology.

We thank the reviewer for their comments and assessment of this work.

However, some major concerns must be addressed before publication. The main ones are:

- The authors refer to the importance of this stem-cell intrinsic properties and the migratory assortment of progenitors to understand planarian regeneration, which is a challenging process because of the arbitrariness of its start point. However, the study is almost focused in the analysis of planarians during homeostasis, while regeneration is superficially studied.

We have now included additional analyses and quantification of specialized neoblasts at wound sites (Fig. 5,, Supplementary Fig. 28-30). Using MERFISH, we mapped the positions of specialized neoblasts at anterior- and posterior-facing wounds during regeneration at 72hpa and 96hpa, respectively, as well as during sagittal regeneration (Fig. 5b,f, Supplementary Fig. 28). We found that specialized neoblasts are highly intermingled in all of these contexts of regeneration, including for regionally specified eye-specialized neoblasts at anterior-facing wounds. To further characterize fate choice in regeneration, we assessed neoblast neighborhoods of specialized neoblasts at anterior-facing wounds 72hpa in pairwise FISH comparisons (new Supplementary Fig. 29). We find that neighborhoods are heterogeneous and often contain specialized neoblasts identified for a different fate, consistent with measurements from uninjured animals. We found that neoblast neighborhoods around eye specialized neoblasts were also highly heterogeneous, with many eye-specialized neoblasts possessing no immediate eye specialized neoblast neighbors (Supplementary Fig. 29a,d). Furthermore, we measured the distances of identified specialized neoblasts at wound sites to different mature tissues, including their respective target tissue in sagittal blastemas, and found that specialized neoblasts were located similar or closer to other tissues than their target tissue and much closer to parenchymal cell types than to target mature tissues in both anterior and posterior regeneration (Supplementary Fig. 30). Taken together, these results indicate that fate choice in regeneration occurs in a spatially heterogeneous manner and does not occur directly adjacent to newly forming target tissues.

-The conclusions reached seem not to consider objectively all the data obtained or all the data that could be easily obtained, since 1) some tendency to find self neoblasts as neighbors is detected, but it is obviated

Our model does not preclude the existence of same-class specialized neoblast neighbors, and in fact, would be expected in the model. For example, in cases consisting of a highly abundant class (e.g. epidermal specialized neoblasts) versus a less abundant class (e.g. muscle specialized neoblasts), it would be expected that self-class neighbors would be present at higher frequency than different-class specialized neoblast neighbors (e.g. high epidermal specialized neoblast neighborhood composition and nearest neighbor proportion from the perspective of epidermal specialized neoblasts). All observed cases demonstrating a significantly higher self-class neighborhood composition and nearest neighbor proportion were comparisons consisting of a highly abundant specialized class compared to a less abundant class from the perspective of the more abundant

specialized neoblast class (Supplementary Fig. 22). If neoblast fates are the result of random sampling, based on the overall frequency of a class in the population, then any given neoblast neighbor has a higher chance of being an abundant class (self or not) than a rare class. Furthermore, in these cases, the immediate neoblast neighborhoods around these neoblasts were heterogeneous, containing specialized neoblasts of different fates. We have clarified the text to better explain this phenomenon.

There were some cases where the nearest neighbor identities of tested neoblast classes were biased toward the self-class as compared to predictions from random simulations, but even in these cases their immediate spatial neighborhoods exhibited extensive heterogeneity.

and 2) regional specific neoblast as the ones corresponding to the eyes are not systematically analyzed. On the contrary, muscle, epidermal and intestinal neoblasts are the ones analyzed, but they correspond to tissues that are all around the planarian body.

We expanded our analysis to include the regionally specified eye and pharynx specialized neoblasts, as suggested by the reviewer. Using whole-mount FISH, we assessed the distance of these populations relative to their mature target tissues and other differentiated tissues, including the intestine, epidermis, and central nervous system (brain and ventral nerve cords) which were detectable by DAPI staining and anatomical position. We found that like other tested fates, eye and pharynx-specialized neoblasts were located closer to different mature tissues than to their target tissue (Fig 3h,i).

Additionally, we assessed the neoblast neighborhoods around eye and pharyngeal specialized neoblasts and found that neighborhood composition was highly heterogeneous and often included specialized neoblasts of different fates (Fig. 3j,k, Supplementary Fig. 24). In many cases, eye-specialized neoblasts had no immediate eye-specialized neoblast neighbors. These results demonstrate that fate specification is spatially intermingled as a general phenomenon, including for regional progenitors. Eye and pharynx-specialized neoblasts are regional, but at the local level exist in highly heterogeneous neighborhoods. Importantly, the regional nature of these specialized neoblasts is a component of the overall model, in which patterning information influences which choices are possible in broad regions, but at the local level fate choice occurs in heterogeneous neighborhoods. To emphasize this aspect of the model, we added new Figure 7b.

Specific issues:

- Pag 3 lines 44 and following. The systematic characterization of FSTFs expression allowed the classification of neoblasts into epidermal, muscle, intestinal, neural, eye, and protonephridial. The ones that not were included to any of these categories were not studied. Since all neoblasts express multiple FSTFs (not always corresponding to the same cell type), which were the criteria followed to classify each neoblast in one category? And to classify them as “unassigned”? Please specify how this categorization was performed, since it is the basis of the following results. It must be explained in the methods in detail.

Thank you for the suggestion. We now include more description of the methodology applied to identify specialized neoblasts in MERFISH experiments (see Methods “Identification and classification of specialized neoblasts by MERFISH”)

- According to Raz et al. 2021, a neoblast can give rise to different fated cells during mitosis. Then, the specialized neoblast identified in this systematic study, can be really considered specialized if it could be that after the next mitosis its fate changes?

In the single-step fate model studied in Raz et al. 2021, a given neoblast can only produce a single type of differentiated daughter in that division, thereby having a restricted fate. In the next division, if one daughter is a neoblast, it could choose a new fate for the next division, reflecting a new fate choice. Importantly, the fate decision of the parent does not need to be inherited by the offspring. Therefore, the single step fate model is compatible with the concept of specialization and fate choice. We modified the text to make this clearer.

- Fig 1c. How is the distance from neoblasts to differentiated tissues calculated? For instances, neural cells or intestine cells are everywhere. Which are the criteria followed? It must be explained in the methods in detail.

We expanded the methods to include a detailed description of the methodology employed. In brief, distances were calculated using the centroid points of identified specialized neoblasts to the nearest nucleus centroid of mature tissues. For neural cell types, we measured the distances of specialized neoblasts to either the cephalic ganglia or ventral nerve cords, whereas in the case of the intestine, distances were measured to the nearest surface of the intestine. These represent the closest possible points in which a new cell may incorporate. (see Methods “Distance measurements of specialized neoblasts to mature tissues by MERFISH”).

- Specialized neoblasts distribution and EdU incorporation experiments lead the authors to suggest that initial fate choices are made everywhere in the planaria and are not related with any positional information. However, postmitotic cells are already found in specific locations related to differentiated cells. How are mitotic cells rearranged to reach their final position? The authors argue that migration must have an important role. However, migration is only demonstrated in irradiation experiments, which is not the context of homeostasis neither regeneration.

If the existence of cell migration cannot be demonstrated in more physiological, non-irradiated conditions, this proposal should remain in the discussion.

Our model posits that neoblast movement in homeostasis is minimal and that the migration of post-mitotic progenitors is the key driver of tissue organization. Neoblast movement has been previously shown to be limited (Guedelhofer and Sánchez Alvarado 2012, Abnave et al. 2017). Several key pieces of prior evidence also support robust migration of post-mitotic cells in homeostatic conditions. Neoblasts are excluded from the pharynx and new pharyngeal cells are specified in the trunk region surrounding the pharynx (Adler et al. 2014, Scimone et al. 2014). Because the pharynx displays robust homeostatic cell turnover and regeneration, post-mitotic specified cells are able to migrate large distances from the site of fate specification into the pharynx. This was shown by multiple studies in which the incorporation of new EdU⁺ or BrdU⁺ cells into the pharynx occurred over time from proliferative cells outside the pharynx itself (LoCascio et al. 2017, Bohr et al. 2021). Similarly, the epidermal lineage has been well studied in planarians and multiple transcriptionally and spatially distinct maturation stages have been identified (Eisenhoffer et al. 2008; Wurtzel et al. 2017). As epidermal progenitors mature, they are observed to migrate away from neoblasts outwards towards the epidermis (Eisenhoffer et al. 2008). Neoblasts are also excluded from the head tip (region anterior to the eyes). Animals pulsed with BrdU demonstrated robust incorporation into cells migrating towards the anterior tip by 6 days post labeling (Newmark and Sánchez Alvarado 2000).

We also expanded our analysis by assessing migration of post-mitotic progenitors in non-irradiated conditions. We pulsed uninjured animals with EdU and assessed the incorporation of new progeny into mature tissues of the anterior head tip, a region devoid of neoblasts; thus, any new incorporation into these tissues must have occurred initially neoblasts, whose post-mitotic progeny then migrated to their target. After pulsing with EdU, we observed robust movement of post-mitotic EdU⁺ cells into the head tip, beyond the region containing neoblasts (Supplementary Fig. 33). We observed incorporation of EdU into anterior-most *cintillo*⁺ sensory neurons and anterior-most *collagen*⁺ muscle cells (Fig. 6d,e). Similarly, we pulsed animals with EdU 4 days post decapitation and assessed incorporation dynamics into the head blastema, again a region excluding neoblasts, at 6dpa. We observed incorporation of EdU into the anterior regions of the brain, indicating that a post-mitotic neural progenitor specified at the wound site had migrated into the anterior regions of the blastema (Fig. 6f).

Using MERFISH, we also identified *smedwi-1* high or *smedwi-1* low FSTF⁺ cells and measured their distances to their respective mature tissues (Supplementary Fig. 32). Previous reports indicate that *smedwi-1* high cells represent neoblast that are actively dividing (S/G2/M phases) whereas *smedwi-1* low cells correspond to non-dividing cells that largely co-express known post-mitotic markers (Zeng et al. 2018; Raz et al. 2021). Previous work has demonstrated that FSTF expression is primarily observed in S/G2/M phase neoblasts and that fate choice can occur within a single cell cycle (Raz et al. 2021). Therefore, the analysis of *smedwi-1* high neoblasts represents detection of neoblasts actively undergoing a fate decision and the approximate spatial location in which that choice is made. We find that *smedwi-1* low FSTF⁺ cells (post-mitotic neoblast descendants that are progenitors) were located significantly closer to target tissues than *smedwi-1* high FSTF⁺ cells, suggesting that *smedwi-1* low post-mitotic progenitors are migrating to their respective target tissues. Together, these data support the model that fate specification can occur at distant sites, and that post-mitotic progenitors migrate to their target destinations. We added these new data and discussion to the manuscript.

- The authors find that adjacent neoblast showed fate specification for tissues from different embryonic germ layers. It is also commented that this is not the case in most animal embryos. However, 1) the existence of neuromuscular precursors in vertebrate and invertebrate embryos has been reported in several studies,

Our discussions were simply intended to point out a common mode of tissue organization during fate specification in many animal embryos, which involves spatially separated germ layers, and that the mode of fate specification observed in this study deviates substantially from that type of physical separation of lineages. As the reviewer points out, neuromesodermal progenitors (NMPs) have been described to contribute to both ectodermal and mesodermal fates within the embryonic tailbud in vertebrates including mouse (Tzouanacou et al. 2009) and zebrafish (Attardi et al. 2018). We included some discussion of NMPs in the Discussion section.

and 2) the comparison of planarians with embryos is interesting, but it should be rather compared with whole-body regenerating animals, as cnidarians or coelms.

There are multipotent progenitors in other animals capable of whole-body regeneration, and it will indeed be of interest to investigate related problems in such organisms in the future. We now include discussion of this point in the context of this work.

In relation to this issue, several single cell sequence analyses have been published on planarian cells in the last few years. The results presented in this study should be discussed according to these data. For instances, are there found cell types that could correspond to precursors of different germ layers, as suggested in the text?

We included more discussion of single cell RNA-sequencing (scRNA-seq) results from planarians throughout the main text. Specialized neoblasts for different fates are robustly identified from scRNA-seq data (van Wolfswinkel et al. 2014, Fincher et al 2018, Plass et al. 2018, Zeng et al. 2018, Raz et al. 2021).

- The authors find muscular, epidermal and excretory specialized neoblasts in all locations and intermingled. However, these tissues are found all along the planarian body. What about region-specific neoblasts?

We expanded our analysis to include eye- and pharynx-specialized neoblasts, which are regionally specified as noted above. We attached the relevant text below

“We have expanded our analysis to include eye and pharynx specialized neoblasts, which are regionally specified. Using whole-mount FISH, we assessed these populations relative to their mature target tissues. We assessed the distance of *ovo*+ eye specialized neoblasts and *foxA*+ pharyngeal specialized neoblasts to their respective target tissues, and to different mature tissues including the intestine, epidermis, and central nervous system (brain and ventral nerve cords) which are detectable by DAPI staining and anatomical position alone. We find that like other tested fates, eye and pharynx specialized neoblasts are located closer to different mature tissues than to their target tissue (new Fig 3h,i).

Additionally, we assessed the neoblast neighborhoods around eye and pharyngeal specialized neoblasts and find that neighborhood composition is highly heterogenous and often include specialized neoblasts of different fate (new Fig. 3j,k, new Supplementary Fig. 24). In many cases, eye specialized neoblasts are found to possess no immediate eye specialized neoblast neighbors. These results are consistent with the prior assessment of other fates, and indicate that fate specification is spatially intermingled as a general phenomenon.”

Specifically, do the authors find eye neoblasts in the prepharyngeal region of an intact animal? An in the postpharyngeal? And in A or P clones of a sublethal-irradiated animal? And what about regeneration? Do the authors find eye neoblasts in A blastemas and also in P blastemas? Without this specific analysis the conclusions reached in the present study are not supported.

Eye-specialized neoblasts are only found within the region posterior to the eyes but anterior to the pharynx in intact animals (Lapan and Reddien 2012, LoCascio et al. 2017, Atabay et al. 2018). Eye specialized neoblasts

are observed in prepharyngeal regions of intact animals (Fig. 3a,c, new Fig. 3f, new Supplementary Fig. 3a). Because of the sparsity of eye-specialized neoblasts (with only ~10-20 detectable cells per homeostatic animal), no eye specialized neoblasts were detected in analyzed small 4-8 cell colonies after sublethal irradiation (also, colonies are ventral biased and eye specialized neoblast are primarily dorsal). Eye-specialized neoblasts are only found at anterior-facing blastemas and not at posterior-facing blastemas (Lapan and Reddien 2011, 2012). We expanded our analysis to include eye-specialized neoblasts at anterior-facing wound sites (new Fig. 5b, new Supplementary Fig. 28a, new Supplementary Fig. 29b,c). We find that the neoblast neighborhoods of eye-specialized neoblasts are highly heterogeneous and composed primarily of specialized neoblasts of a different class or unidentified neoblasts in homeostasis and regeneration (new Fig. 3j, new Fig. 5e, new Supplementary Fig. 29d).

Additional studies that should be performed in this line: if A and P clones in sublethal irradiated animals are analyzed, are there differences in the composition of the clones related with the AP position? This kind of analysis should be performed, since the finding of some differences would mean that positional cues are in fact important to specify cell fate during mitosis.

Our model does not intend to exclude the existence of important positional cues in fate specification; indeed, we know this to be the case: eye specialized and pharynx specialized neoblasts demonstrate the key role of position in fate choice demonstrated by their regionally constrained specification zones. We expanded our model to make this aspect of fate choice regulation clearer (Fig. 7b). Similarly, we find in our data that intestinal specialized neoblasts show some positional bias to the intestine, indicating that position may also play a role in the specification of this fate. However, our current homeostasis and regeneration data indicate that whereas position may broadly influence fate specification options, individual neoblast neighborhoods are highly heterogeneous, with different fates frequently specified in adjacent neoblasts.

Colony analysis presents inherent limitations that restrict the scope of fate distribution analysis. To generate a subtotally irradiated animal with colonies of sufficient sparsity to be confidently identified as distinct colonies, animals must be irradiated at high doses that result in total clearance for most animals in a sample. Due to the technical limitations of wholemount FISH, at most 2 specialized neoblast classes can be assessed in early neoblast colonies after subtotal irradiation. We did not observe any AP spatial bias in identified colonies in this work or previous work (Raz et al. 2021). Continued technical advancement of colony analysis methods will allow us to revisit these analyses in further detail. In this work, specialized neoblasts of multiple fates were broadly distributed along the AP axis and could be observed in both anterior- and posterior-facing wounds.

- In the analysis of Fig3D and Supp 21, the authors identify the neighbors for a given neoblast and conclude that this data supports their previous data and a model in which fate specification occurs in a highly intermingled “salt-and-pepper” distribution. However, looking at the data, it is true than in some cases non-self neoblasts are the more represented neighbors, but in others are the self neoblasts. These cases should be also commented and included in the discussion of the results. Furthermore, the analysis of regional neoblasts as the ones corresponding to the eyes or to the pharynx region should be included in the analysis.

We have added more text clarification to describe these findings. As noted above, different specialized neoblast classes are not present at uniform proportions within the animal, with some fates such as the epidermis composing a large fraction of specialized neoblasts (van Wolfswinkel et al. 2014). Therefore, comparisons containing epidermal specialized neoblasts, for example, would be expected to produce a higher proportion of epidermal neighbors than other specialized classes by random sampling. We have also expanded analyses to include quantification of eye and pharynx specialized neoblast neighborhoods (Fig. 3h-k, Supplementary Fig. 24).

Regarding this data represented in the graphs of Fig3d-e and Supp 21, a better explanation of the graphs or an improvement should be performed to make it easy their interpretation.

We modified the figure legend descriptions to improve the description of the data.

As introduced in the manuscript, the biological question behind this study is to understand how stem cells decide their fate during regeneration. However, almost all the analysis are performed in homeostatic animals. The same analysis performed in intact animals should be performed in regenerating blastemas. Are the specialized neoblast closer to their corresponding differentiated structures? Are neural fated neoblasts closed to the regenerating nerve cords? Are intestinal fated neoblasts closed to the regenerating intestine? Are self neoblasts closer between them than to non-self neoblasts? As already pointed out, are eye neoblasts present in P blastemas?

The authors have the tools to answer this key question, which was the initial interest, according to the introductory lines.

We expanded our analyses to include specialized neoblasts at wound sites. (Fig. 5, Supplementary Fig. 28-30). We found that specialized neoblasts are highly intermingled in anterior, posterior, and sagittal regeneration and neoblast neighborhoods of specialized neoblasts are heterogeneous (Fig. 5b-f, Supplementary Fig. 28,29). We also found that specialized neoblasts were located at similar or closer distances to other tissues than to their corresponding mature tissue (Fig. 5g, Supplementary Fig. 30). We also examined distance between specialized neoblasts and target and non-target tissues in sagittal blastemas, and were similarly distant. Eye-specialized neoblasts also demonstrated highly heterogeneous neighborhoods at anterior-facing wounds (Fig. 5e, Supplementary Fig. 29). Eye-specialized neoblasts are not present in posterior-facing wounds and within the tail during sagittal regeneration.

- What is the point of the shielded irradiated planarians experiment? The migration of neoblasts in these conditions has already been reported in different studies (Dr. Aziz lab). Furthermore, as already pointed out, if the authors would like to demonstrate the role of migration in positioning differentiated cells, it should be demonstrated in regenerating or intact animals, but not in a context in which the population of neoblasts is dramatically decreased and migration is the only way of survival. It has been demonstrated in several animal models that cell behaviors is not comparable between physiological and extreme conditions of cell depletion.

Whereas previous studies (Abnave et al. 2017; Guedelhofer and Alvarado 2012) demonstrated that post-mitotic progenitors for the epidermal lineage (early *prog-1+* and late *agat-1+*) could migrate farther that neoblasts from shielded regions after lethal doses of irradiation. However, successful targeting and incorporation of these migratory cells into mature tissue was not determined in these studies, and only one fate was assessed. In our studies, we expanded these results through the application of EdU labeling, which allows for the identification of post-mitotic cell types born from surviving neoblasts in the shielded region throughout the animal. In our study, we identify clear incorporation of EdU+ cells into the anterior epidermis, far from the shielded region and regions containing neoblasts, as well as incorporation of EdU into the cephalic ganglia by *chat* co-staining.

We also expanded our analysis by assessing migration in non-irradiated homeostatic and regeneration conditions, as suggested. We pulsed uninjured animals with EdU and assessed the incorporation of new progeny into mature tissues of the anterior head tip, a region devoid of neoblasts; thus, any new incorporation into these tissues must have occurred initially within distant neoblasts, then migrated to their target. After pulsing with EdU, we observed robust movement of post-mitotic EdU+ cells into the head tip, beyond the region containing neoblasts (Supplementary Fig. 33a,b). We observed incorporation of EdU into anterior-most *cintillo+* sensory neurons and anterior-most *collagen+* muscle cells (Figure 6d,e). Similarly, we pulsed animals with EdU 4 days post decapitation and assessed incorporation into the blastema, again a region excluding neoblasts, at 6dpa. We observed incorporation of EdU into the anterior regions of the brain, indicating that a neural cell specified at the wound site had migrated into the anterior regions of the blastema (Fig. 6f, Supplementary Fig. 33d). Together, these data all support the model that fate specification can occur at distant sites, and that post-mitotic progenitors migrate to their target destinations.

Additional issues:

- From supp 7 to 10, Fig 3B, 3F, supp 23... please show the region to which the cells correspond.

We specified the regions in corresponding figure legends.

- Fig 1A, muscle and pigment cells are difficult to be distinguished.

We adjusted the color to better distinguish between these two cell types.

- Movies are not named as S1, etc...

We renamed files accordingly.

- Page 6 line 4- “subtotal” should be “sublethal”.

We stayed with subtotal to refer to the fact that not all neoblasts are removed in all animals, because some animals will die at the dose used, and for consistency with usage in Raz et al. 2021.

- It is hard to understand this manuscript for the general reader. Basic planarian biology should be included: the cellular and molecular basis of planarian regeneration, the stages of regeneration, what is a sublethal irradiation...

Thank you for this comment as we want the text to be accessible to a broad audience; we added more descriptive detail throughout the text.

- It is difficult to follow the results in the current format. Figures should contain information about the region analyzed, and explained in more detail.

We included cartoons and descriptions to describe assessed regions.

- Supp 24. Circles must be of the same thickness (all thinner).

We reduced the thickness of the circles in the figure.

- Pag4-line 8 “Supplementary Fig. a,b”- number is missing.

We modified the text accordingly.

- The authors found that specialized neoblasts were closer to parenchymal cells, and in several cases also with intestinal cells. A possible explanation could be the role as niches of these tissues. This issue could be discussed.

The same possibility crossed our minds some time ago, and in other work in the lab we ablated the parenchymal cells without revealing an obvious niche-like role so far (but it is not completely ruled out yet). Thus, we don't want to go too far in speculation about this in the text; nonetheless, we highlight this interesting physical association in the text a bit more now.

- The model proposed in Fig 4d is too simplistic. A more elaborated model, in line with a more elaborated discussion, would enrich the manuscript.

We expanded the model (Fig. 7b) to include more detail about the regional influence of fate choice.

REVIEWERS' COMMENTS

Reviewer #1 (Remarks to the Author):

In this paper, Park, Owusu-Boaitey and colleagues have explored whether planarian stem cells are specified to various fates needed nearby to contribute to local cell production. Surprisingly, the authors find that planarian stem cells give rise to different fates (even fates of different germ layers) in a way that is agnostic to cell location in the animal. Instead, the authors argue, stem cell progeny migrate to the sites where they are needed well after specification has occurred. This work is important and impactful, raising new questions and incorporating new methods to advance the field.

Park, Owusu-Boaitey and colleagues have completed a thorough and rigorous revision of this manuscript for its second submission. I have only minor questions that the authors can choose to address if they would like.

1. It could be valuable to add a note in the discussion about the "chaotic" embryonic development of planarians. It seems likely that the randomized spatial specification pattern would also lend itself well to embryonic development in this organism.
2. It is very challenging for the reader to determine which probes are used for data in each figure (especially for pools). It would be great to add in the figure legends the probes there were used for each figure ("CNS = gene1, gene2, gene3"). I found myself having to cross-check and refer back frequently, leading to some confusion as I tried to interpret the data for myself.
3. The arrowheads in Supp. Fig. 4 seem to be pointing at nothing. The authors can check to make sure that they didn't shift in image processing.
4. In Fig. 2, "Non-self" would seem to be clearer than "No ID."

Reviewer #2 (Remarks to the Author):

My concerns have been addressed in the revised version of the manuscript. I fully support the publication of this revised version in Nature Communications.

Reviewer #3 (Remarks to the Author):

The authors have included in the revised version new experiments and data that answer the main concerns raised. The rebuttal letter is also argued and convincing. I recommend its publication in Nature Communications.

The only suggestion is to improve the final model, which is not clearly presented in Figure 7 neither in the discussion. In Figure 7a, the model suggests that the mechanism to organize tissues from the intermingled neoblasts is migratory sorting. However, this is not demonstrated neither supported by any data in this or other publications. It is only a model, but it is an important aspect. Other mechanisms like the effect that differentiated tissues can have in deciding the final fate of nearby cells must be considered. Furthermore, the message in Figure 7b is not clear at all. According to the figure legend it is a “model depicting regional influence on neoblast fate choice across the anteroposterior axis”. However, this is not understood looking at the image. And also, what about the DV axis? What is the role of PCGs here, if any?

REVIEWERS' COMMENTS

Reviewer #1 (Remarks to the Author):

In this paper, Park, Owusu-Boaitey and colleagues have explored whether planarian stem cells are specified to various fates needed nearby to contribute to local cell production. Surprisingly, the authors find that planarian stem cells give rise to different fates (even fates of different germ layers) in a way that is agnostic to cell location in the animal. Instead, the authors argue, stem cell progeny migrate to the sites where they are needed well after specification has occurred. This work is important and impactful, raising new questions and incorporating new methods to advance the field.

Park, Owusu-Boaitey and colleagues have completed a thorough and rigorous revision of this manuscript for its second submission. I have only minor questions that the authors can choose to address if they would like.

We thank the reviewer for their constructive feedback and support of this work.

1. It could be valuable to add a note in the discussion about the "chaotic" embryonic development of planarians. It seems likely that the randomized spatial specification pattern would also lend itself well to embryonic development in this organism.

We agree with the reviewer that this is a very interesting topic for future study. While early embryonic development is indeed seemingly spatially random, we chose not to discuss planarian embryonic development as we did not want to overly speculate given no information about planarian embryonic development is provided in this work.

2. It is very challenging for the reader to determine which probes are used for data in each figure (especially for pools). It would be great to add in the figure legends the probes there were used for each figure ("CNS = gene1, gene2, gene3"). I found myself having to cross-check and refer back frequently, leading to some confusion as I tried to interpret the data for myself.

We now include a list utilized probes in pools for all FISH experiments in appropriate figure legends. For MERFISH probe pools, given some cases contain >10 gene targets per pool, we refer in the legend to the supplemental table for this information.

3. The arrowheads in Supp. Fig. 4 seem to be pointing at nothing. The authors can check to make sure that they didn't shift in image processing.

The image has been corrected.

4. In Fig. 2, "Non-self" would seem to be clearer than "No ID."

We modified the text to Non-Self for clarity.

Reviewer #2 (Remarks to the Author):

My concerns have been addressed in the revised version of the manuscript. I fully support the publication of this revised version in Nature Communications.

We thank the reviewer for their constructive feedback and support of this work.

Reviewer #3 (Remarks to the Author):

The authors have included in the revised version new experiments and data that answer the main concerns raised. The rebuttal letter is also argued and convincing. I recommend its publication in Nature Communications.

We thank the reviewer for their constructive feedback and support of this work.

The only suggestion is to improve the final model, which is not clearly presented in Figure 7 neither in the discussion. In Figure 7a, the model suggests that the mechanism to organize tissues from the intermingled neoblasts is migratory sorting. However, this is not demonstrated neither supported by any data in this or other publications.

We modified the wording of the text to make prior findings and findings in this work related to migration of post-mitotic neoblast descendant cells clearer. In short, nucleotide analogs label neoblasts (the only dividing cells) and this marks neoblast descendant cells permanently thereafter, allowing visualization of their localization relative to neoblasts. BrdU labeling showed neoblast descendants rapidly end up in head tips, where neoblasts themselves are absent (Newmark and Alvarado 2000). Similarly, neoblast descendants end up throughout the pharynx, despite this organ lacking neoblasts and pharynx progenitors being specified outside of the pharynx (LoCascio et al. 2017, Bohr et al. 2021). Spatially distinct maturation states can be observed in epidermal and eye progenitors indicating that these progenitors can be specified distantly and migrate to their target tissues as they progress in their gene expression state towards differentiation (Eisenhoffer et al. 2008, Lapan and Reddien 2012, Wurtzel et al. 2017). Next, regenerating eyes are observed to increase proportionally to the number of eye progenitors present in eye trails, indicating that distant progenitors can productively incorporate into eye tissue (Lapan and Reddien 2011). In this experiment, animals were lethally irradiated to prevent new eye cell production (e.g. production of new eye cells close to the eye). Furthermore, early *prog-1+* epidermal progenitors are observed to migrate away from *smedwi-1+* neoblasts in shielded irradiation experiments (Abnave et al. 2017).

In this work, we observed that differentiating *smedwi-1* low/*FSTF+* cells are positioned closer to target tissues than *smedwi-1* high/*FSTF+* neoblasts, consistent with movement of early post-mitotic progeny. Additionally, we observe rapid change in distribution of *EdU+smedwi-1-* neoblast progeny into the anterior head tip, a region devoid of neoblasts, and formation of neural and muscle cell types in homeostasis and regeneration far from neoblasts that specify these cells. In shielded irradiation experiments, we observed incorporation of neoblast descendant cells (labeled with *EdU*) into neural and epidermal tissue far beyond the region

containing remaining neoblasts. Taken together, these results provide substantial support for a model in which post-mitotic progeny in planarians undergo migration to target tissues.

It is only a model, but it is an important aspect. Other mechanisms like the effect that differentiated tissues can have in deciding the final fate of nearby cells must be considered. We agree that it remains possible that differentiated cells could influence final fate of post-mitotic neoblast descendants and included some text in the discussion to this effect. However, if differentiated tissues were determining the pattern of fate choice in neoblasts, a greater degree of spatial structure of neoblast neighborhood composition and distances to target tissues would be expected than observed. Similarly, no evidence of hybrid-state neoblasts or post-mitotic progenitors are observed in single-cell sequencing data which would be expected if such progenitors exhibited fate-switching (Fincher et al. 2018). In previous work, distant eye progenitor cells were shown to contribute to overall eye growth in regeneration (Lapan and Reddien 2011), and spatially distinct maturation stages have been observed for eye and epidermal progenitors as they differentiate (Eisenhoffer et al. 2008, Lapan and Reddien 2012, Wurtzel et al. 2017), indicating fate choice can occur far from target tissues. Finally, when an eye was transplanted outside of the normal eye progenitor specification zone, it could not be maintained through cell turnover, indicating the eye could not influence the fate of local neoblasts or post-mitotic neoblast progeny into an eye fate (Atabay et al. 2018).

Furthermore, the message in Figure 7b is not clear at all. According to the figure legend it is a “model depicting regional influence on neoblast fate choice across the anteroposterior axis”. However, this is not understood looking at the image. And also, what about the DV axis? What is the role of PCGs here, if any?

We added additional detail and description to Figure 7b. We labeled with an asterisk in the model figure cells that choose a region-appropriate fate, but still in heterogeneous neighborhood. Whereas the role of PCGs were not explicitly explored in this work, several pieces of evidence suggest the influence of PCGs on fate choice. For example, the DV axis is known to influence epidermal fate, and we have included this detail in the figure legend (Wurtzel et al. 2017). Similarly, specification of eye and pharynx fates are known to occur regionally along the AP axis. In conditions where AP patterning is disrupted (e.g. *B-catenin* RNAi), ectopic eye formation is observed in posterior facing wounds in regeneration and along the AP axis in homeostasis (Gurley et al. 2008, Peterson and Reddien 2008). Under *notum* RNAi, ectopic eyes anterior to the pre-existing eyes are produced – under this condition, the spatial distribution of *ovo+* eye progenitors is also observed to shift anteriorly (Hill and Petersen 2018). However, although eye specification is regional and may be influenced by changes in PCG expression, our work demonstrates that eye fates can be specified far from the eyes and local neoblast neighborhoods around eye specialized neoblasts are heterogeneous and can contain neoblasts specifying different fates.